# IDS-AGENT: AN LLM AGENT FOR EXPLAINABLE INTRUSION DETECTION IN IOT NETWORKS

## ABSTRACT

Emerging threats to IoT networks have accelerated the development of intrusion detection systems (IDSs), characterized by a shift from traditional approaches based on attack signatures or anomaly detection to approaches based on machine learning (ML). However, current ML-based IDSs often lack result explanations and struggle to address zero-day attacks due to their fixed output label space. In this paper, we propose `IDS-Agent`, the first IDS based on an AI agent powered by large language models (LLMs). For each input network traffic and a detection request from the user, `IDS-Agent` predicts whether the traffic is benign or being attacked, with an explanation of the prediction results. The workflow of `IDS-Agent` involves iterative reasoning by a core LLM over the observation and action generation informed by the reasoning and retrieved knowledge. The action space of `IDS-Agent` includes data extraction and preprocessing, classification, knowledge, and memory retrieval, and results aggregation – these actions will be executed using abundant tools, mostly specialized for IDS. Furthermore, the `IDS-Agent` is equipped with a memory and knowledge base that retains information from current and previous sessions, along with IDS-related documents, enhancing its reasoning and action generation capabilities. The system prompts of `IDS-Agent` can be easily customized to adjust detection sensitivity or identify previously unknown types of attacks. In our experiments, we demonstrate the strong detection capabilities of `IDS-Agent` compared with ML-based IDSs and an IDS based on LLM with prompt engineering. `IDS-Agent` outperforms these SOTA baselines on the ACI-IoT and CIC-IoT benchmarks, with 0.97 and 0.75 detection F1 scores, respectively. `IDS-Agent` also achieves a recall of 0.61 in detecting zero-day attacks, outperforming previous approaches specially designed for this task.

## 1 INTRODUCTION

In recent years, the Internet of Things (IoT) has emerged as a transformative technology, increasingly adopted across a wide range of applications (Chataut et al., 2023). Alongside its rapid development, security concerns have arisen within IoT networks due to the typically large number of devices with potential trustworthiness issues (Alghofaili & Rassam, 2022). Therefore, the deployment of Intrusion Detection Systems (IDSs) has become essential, as they play a critical role in monitoring network traffic and identifying malicious activities (Khraisat et al., 2019).

Many traditional IDS employ signature-based methods, which rely on signatures of known attacks stored in databases (Ioulianou et al., 2018; Nagaraju et al., 2021; AlYousef & Abdelmajeed, 2019). They suffer from high false negative rates when variations in attack methodologies do not exactly match the existing signatures. Alternatively, IDS can also be designed as an anomaly detector to identify distributional deviations from normal traffic (Khraisat et al., 2018; Jia et al., 2021). However, defining the normal behavior in a network can be challenging, especially in diverse and dynamic environments where normal activity can change over time. Machine learning (ML)-based IDS was then proposed to address these problems by leveraging the representation power of ML models, such as deep neural networks (DNNs), to capture complex attack patterns from extensive training data (Maseer et al., 2021). However, ML-based IDS still face limitations due to constraints in the model architecture and training data, which hinder their detection capabilities, especially when

addressing zero-day attacks (Ullah et al., 2019). Furthermore, the detection results from ML-based IDS often lack clear explanations, which can diminish their credibility, particularly in safety-critical IoT scenarios where understanding the rationale behind alerts is crucial (Catarinucci et al., 2015; Ahmad & Zhang, 2021).

Recently, AI agents empowered by large language models (LLMs) have been created to handle complicated tasks in various application domains (Yu et al., 2023; Abbasian et al., 2024; Shi et al., 2024; Li et al., 2024; Deng et al., 2023; Gur et al., 2024; Zheng et al., 2024). These agents are characterized by their integrated capabilities to observe the environment, generate a reasonable plan, and then take appropriate actions according to the user's requests. Typically, LLM agents are equipped with a knowledge/memory base and a toolbox that includes tools such as local functions and third-party APIs. They utilize one or more LLMs for knowledge/memory-based reasoning and planning, including the selection of the most effective tools to employ. The intelligence of LLM agents in reasoning (as well as analysis and criticism (Shinn et al., 2023)) makes them both powerful operators and effective intermediaries between task execution and human users.

In this paper, we propose `IDS-Agent`, the first LLM agent designed for intrusion detection, featuring capabilities for explanation, customization, and adaptation to zero-day attacks. `IDS-Agent` takes an input request for intrusion detection with a target raw data flow, and outputs detection results with a detailed explanation. The agent adopts a reasoning-followed-by-action pipeline ( (Yao et al., 2023)) with a specialized action space for intrusion detection. Specifically, knowledge-enabled reasoning based on long-term memory from previous sessions is performed by the core LLM of the agent to decide the optimal tools (and their settings/parameters) for data extraction, preprocessing, classification, and results aggregation.

Compared with existing ML-based IDSs, `IDS-Agent` achieves a stronger detection performance and better interpretability by harnessing the power of multiple ML models and external knowledge in a comprehensive way. It aggregates the classification results from multiple ML models by prompting an LLM with the top-k label predictions and their confidence scores for each model. The prompt also includes external knowledge (e.g. regarding particular attack types) obtained by calling a search engine and additional instructions, for example, to customize detection sensitivity or to reveal new attack types. The LLM is instructed to produce structured outputs, including detection results and an explanation. Our main technical contributions are summarized below:

- We propose `IDS-Agent`, the first LLM-powered agent for intrusion detection, featured by its capabilities of results explanation, detection customization, and revealing of zero-day attacks.
- We propose a reasoning-followed-by-action pipeline for `IDS-Agent` with an action space and toolbox specialized for network traffic processing and intrusion detection results aggregation.
- We demonstrate the effectiveness of `IDS-Agent` on two IDS benchmarks, ACI-IoT'23 and CIC-IoT'23. `IDS-Agent` achieves higher detection accuracy compared with the latest LLM-based method, various ML models, and the majority voting classifier.
- Experiments show that when classifiers produce discrepancy predictions, `IDS-Agent` can utilize inherent and external knowledge to help the decision-making. `IDS-Agent` also demonstrates clearly better performance than existing approaches in detecting zero-day attacks. Moreover, we find that `IDS-Agent` effectively follows the sensitivity instructions without requiring expert intervention or additional tuning.

## 2 RELATED WORK

**Conventional IDS** An Intrusion Detection System (IDS) is designed to detect malicious activities on computer systems, helping to ensure system security (Khraisat et al., 2019). IDSs are generally classified into two main types: Signature-based Intrusion Detection Systems (SIDS) and Anomaly-based Intrusion Detection Systems (AIDS). SIDS relies on pattern-matching techniques to identify known attacks (Ioulianou et al., 2018; Nagaraju et al., 2021; AlYousef & Abdelmajeed, 2019). However, the rise in zero-day attacks has increasingly diminished the effectiveness of SIDS, as these new attacks lack existing signatures (Ullah et al., 2019). In contrast, AIDS constructs a model of normal system behavior using machine learning, statistical, or knowledge-based techniques. Any significant deviation from the constructed model is flagged as an anomaly, potentially indicating an intrusion (Khraisat et al., 2018; Jia et al., 2021). Knowledge-based AIDS requires creating a

knowledge base that reflects the legitimate traffic profile by using techniques such as description language (Studnia et al., 2018) and expert system (Kim et al., 2014). However, in a dynamically changing environment, a time-consuming regular update on the knowledge base is needed. Compared with these conventional IDS, our `IDS-Agent` can adopt a search engine to obtain up-to-date knowledge and can handle zero-day attacks, as shown by our experiments.

**ML-based IDS** Many machine learning models, such as MLP (Bajaj & Arora, 2013), KNN (Li et al., 2014), Decision Tree (Guezzaz et al., 2021), SVM (Mohammadi et al., 2021), have been explored for anomaly-based intrusion detection. For IoT intrusion detection, Verma & Ranga (2020) conducted a comprehensive comparison of ensemble and individual classifiers, including Random Forest (RF), AdaBoost (AB), and Gradient Boosted Machines (GBM). Roy et al. (2022) proposed a lightweight IDS model utilizing machine learning to detect cyber-attacks and anomalies in resource-constrained IoT systems. Davis et al. (2024) advanced this line of research by applying a quantum-annealing approach for feature selection in IoT intrusion detection. Compared with ML-based IDSs, our `IDS-Agent` not only achieves better empirical performance but also provides a detailed explanation of each intrusion detection result.

**LLM-based IDS** Large language models (LLMs), especially generative pre-trained commercial transformers, like GPTs, have recently demonstrated outstanding ability in information comprehension and reasoning tasks. This has motivated some studies in applying LLMs to abnormal detection tasks, such as compiler optimization (Gu, 2023) and software vulnerability detection (Guo et al., 2024). Zhang et al. (2024) is the first to use LLMs for IDS by employing a straightforward in-context learning approach with GPT-4, which provides it with a few labeled examples. Their method achieved over 90% accuracy on a simple dataset containing only five types of attacks. However, in this paper, we demonstrate that the performance of their method drops significantly when tested on more complex and diverse datasets. Different from this LLM-based IDS, our `IDS-Agent` uses LLM for reasoning, action planning, and tool selection, leading to a huge performance gain on diverse datasets.

# 3 METHOD

## 3.1 OVERVIEW

`IDS-Agent` is designed to produce a prediction result with an explanation for each user request for IoT traffic inference, i.e., to determine if the traffic is benign or belongs to any particular attack type. We also allow `IDS-Agent` to handle requests for customized detection sensitivities or to detect new attack categories from the given IoT traffic flow. `IDS-Agent` is equipped with a) an abundant toolbox containing special IDS tools such as ML models for classification and general tools such as search engines to retrieve external knowledge and b) a memory and knowledge base storing the current session information, long-term memory from previous sessions, and supportive documents. These tools, memory, and external knowledge will be integrated to guide the decision-making of `IDS-Agent` in a structured manner, as detailed in the sequel.

## 3.2 PIPELINE OF IDS-AGENT

The pipeline of our `IDS-Agent` is inspired by the ReAct agent (Yao et al., 2023). The user request is fulfilled by executing a sequence of action steps $\{a_1, a_2, \cdots\}$, where each action step is generated by a core LLM based on previous reasoning and observations. For any input user request with a specification of the traffic flow to be inferred, an initial observation $o_0$ is constructed by concatenating the user request with a system prompt, including a description for each available tool. This initial observation serves as the context for the agent to understand the task, facilitating subsequent reasoning and action generation.

Specifically, `IDS-Agent` iterates over the following three steps after the construction of $o_0$:
**1) Reasoning:** The core LLM generates a thought (in plain text) about the next action by $r_i = \text{LLM}(s_i)$, where $s_i = \{o_0, \{r_1, a_1, o_1\}, \cdots, \{r_{i-1}, a_{i-1}, o_{i-1}\}\}$ is the short-term memory of the current session up to the $(i-1)$-th iteration (with $s_1 = \{o_0\}$). Reasoning can optionally adopt long-term memory from previous sessions for in-context demonstration.
**2) Action generation:** The action is generated based on the reasoning/thought by $a_i = \text{LLM}(r_i, s_i)$. Notably, each action we generate is a structured JSON file containing an *action name* and an *action input*. The action input consists of the name of the tool(s) to be used and the associated settings or

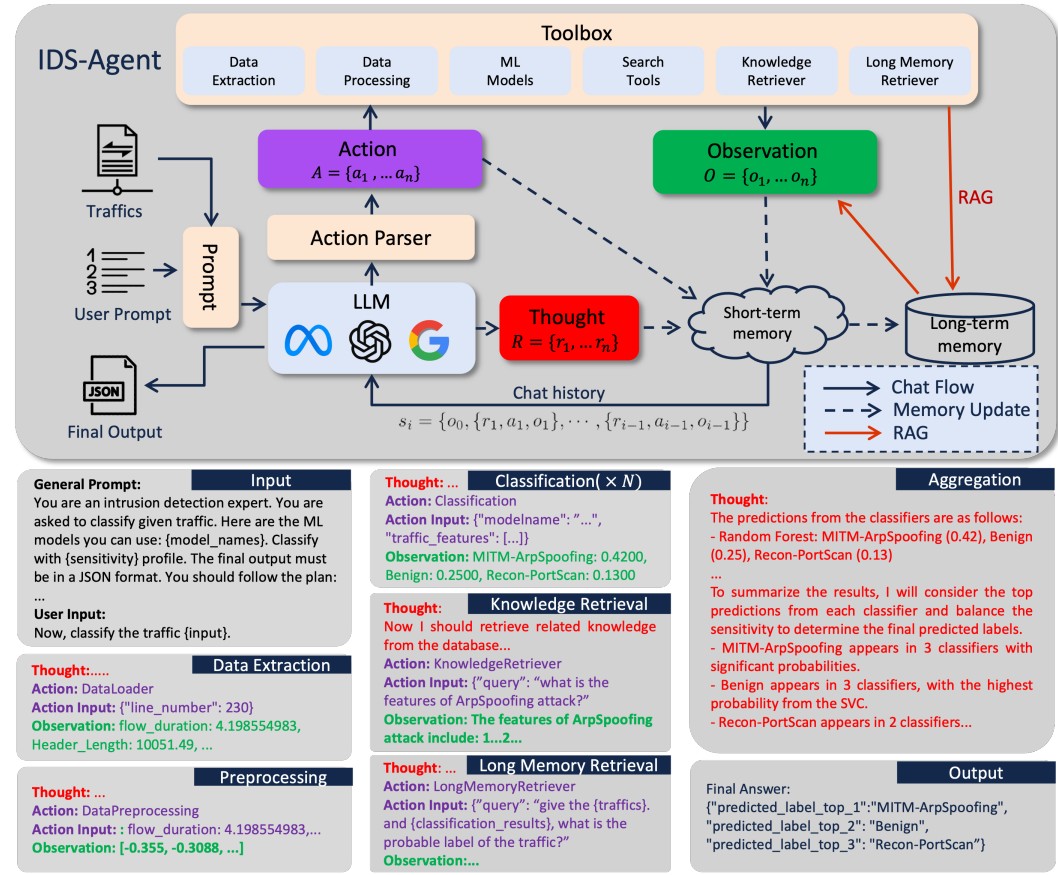

Figure 1: Overview of `IDS-Agent`. **Top** (framework of `IDS-Agent`): `IDS-Agent` adopts a core LLM to generate thoughts (i.e. reasoning) and actions based on input traffic and user prompt. It is equipped with a toolbox for action execution and a memory base for knowledge retrieval. `IDS-Agent` iteratively conducts thought generation, action generation and execution, and observation update. **Bottom** (an example of reasoning trace): in this example, several classifiers are adopted by the agent, with same number of classifiers predicting the input traffic as 'Benign' and 'MITM-ArpSpoofing'. Based on this observation, `IDS-Agent` decides to perform 'Knowledge Retrieval' and 'Memory Retrival', and finally aggregate these observations, which leads to correct attack detection.

parameters of each tool. Such a structured generation of the action allows its efficient and accurate execution using the specified tools.

**3) Observation update:** After executing the generated action $a_i$, we obtain a new observation $o_i$ by converting the outputs of the tool(s) into plain text.

The iterations terminate when the observation is updated by a 'final answer' headline followed by a JSON file. This JSON file, which encapsulates the final prediction on the traffic data and related analysis and explanation, will be the output of `IDS-Agent`.

### 3.3 ACTION SPACE AND TOOL DESIGN

Our `IDS-Agent` is designed with a comprehensive action space, allowing it to handle various tasks in the pipeline of data processing and classification through iterative reasoning and execution. The action space includes the key actions described below.

**Data Extraction**: The goal is to accurately extract network traffic records $x$ specified in the user request from the dataset for further analysis. We design the data extraction tool as a function that takes the given line number or the flow ID as the input and outputs a structured traffic sample.

**Preprocessing**: The goal is to clean, normalize, and transform the extracted data into a standard format for classification. Our preprocessing tools are designed as functions for diverse data analysis operations, including feature scaling, data encoding, handling missing values, and selecting important features for classification.

**Classification**: This action applies machine learning models to the preprocessed data to obtain classification results, i.e., to predict whether the traffic is benign or falling into any malicious category. The inputs to the classification tool include the preprocessed traffic features and a classifier, while the outputs include the top-k labels and their corresponding confidence scores. Note that our classification tool is more than a classifier; it advances ML-based IDS by adapting to diverse model types and incorporating more information from the classification results into the inference procedure. The types of classifiers used include Random Forest, SVM, MLP, Decision Tree, KNN, etc. It is important to note that the classification toolbox is extensible. In real-world deployments, users can easily add new classifiers to the toolbox without the need to fine-tune the LLM. This flexibility allows for easy updates and adaptation to new attack patterns or changes in the network environment. Moreover, the classifiers can also be open-sourced models trained by third parties (callable through APIs), or models locally trained based on the data collected by the user.

**Knowledge Retrieval**: This action aims to obtain knowledge regarding the particular types of attacks predicted by the classifiers. The knowledge can be either external knowledge and retrieved by calling search engines such as Google and Wikipedia API, or stored knowledge base or predefined rules and retrieved by RAG. The retrieved knowledge will be used to guide the aggregation step.

**Long Memory Retrieval**: Long Memory carrying information from previous sessions (discussed in Sec. 3.4). We add an instruction in the system prompt in $o_0$ such that long-term memory retrieval will be activated (through reasoning).

**Aggregation**: This action aims to comprehensively integrate the results from multiple steps of classification action (based on different classifiers) to generate a structured final inference decision. The core of the aggregation tool is an LLM where the prompt is designed to include 1) the detailed results from the classifiers, 2) the short-term memory, and 3) demonstrative inputs and outputs aggregated by the LLM from previous sessions. Note that the short-term memory also includes the external knowledge previously extracted and the system prompts in $o_0$. This system prompt, as shown in Fig. 3, includes a specification for the detection sensitivity; it can also include an instruction for revealing new attack categories. Compared to naive aggregation, such as majority voting, our method incorporates more information to resolve any discrepancies between different model outputs in a more comprehensive way.

### 3.4 MEMORY AND KNOWLEDGE BASE

The memory and knowledge base of `IDS-Agent` stores 1) the short-term memory, 2) the long-term memory, and 3) supportive documents for IDS.

**Short-term Memory (STR).** The STR, as described in Sec. 3.2, includes the historical reasoning trace, actions, and observations of the current session in a structured format, and is renewed after each observation update. The major goal of the STM here is to track the agent's iterative reasoning process and ensure consistency between steps in real-time.

**Long-term Memory (LTM).** The LTM consists of agent decisions and contextual information from previous use cases (Wang et al., 2024; Zhong et al., 2024). Here, a structured LTM example is defined by $\phi = \{t, x, R, A, O, \hat{y}\}$, where $t$ is a timestamp, $x$ is the feature vector after preprocessing, $R = [r_1, \cdots, r_n]$ is the reasoning trace, $A = [a_1, \cdots, a_n]$ contains all the action steps, $O = [o_1, \cdots, o_n]$ are the observations, and $\hat{y}$ is the final label prediction. The long-term memory base can be initialized by running the agent on a validation dataset. During the inference, only sessions with correct agent decisions will be stored in the long-term memory base. In a real-world intrusion detection scenario, such correctness can be validated by human experts.

In our framework, LTM retrieval provides the agent with additional information while aggregating the results for individual classifiers. Here, we set the LTM retriever input as the current timestamp $t$ and observations from previous data processing and classification actions, denoted by $\tilde{O} = [o_1, \ldots, o_{m-1}]$, where $m$ is an arbitrary iteration where the LTM retrieval kicks in. The retriever obtains the top-k relevant final reasoning based on the weighted sum of timestamp distance and the cosine similarity

between the embedding of $\tilde{O}$ and the observation embeddings $O^{(j)}$ of previous LTM examples $\{\phi^{(1)}, \cdots, \phi^{(L)}\}$. Specifically, we obtain the top-k solutions to

$$\text{argmax}_j[\lambda_1 r(t, t^{(j)}) + \lambda_2 \text{cosim}(\mathcal{E}(\tilde{O}), \mathcal{E}(O^{(j)}))], \tag{1}$$

where $\mathcal{E}(\cdot)$ is the encoder. $r(t, t^{(j)}) = 1 - (t - t^{(j)})/\max_k(t - t^{(k)})$ is the recency of the memory. The equation ensures that both recent observations and content-wise similar observations are considered to address the evolving nature of intrusion data. Then, the observation $o_m$ for the iteration $m$ contains the input-prediction pairs $(x^{(j)}, \hat{y}^{(j)})$ of each retrieved $\phi^{(j)}$. In our experiments, we set $k = 5$ to retrieve the top 5 relevant structured LTM examples.

**External Knowledge.** In addition to the external knowledge obtained by calling search engines, such as Google and Wikipedia, `IDS-Agent` is also equipped with a vector database $\{\psi^{(1)}, \cdots, \psi^{(K)}\}$ containing related research papers and intrusion detection blogs (both parsed into chunks with fixed token length). The retrieval from this knowledge base is similar to the retrieval of LTM. We obtain the top-k solutions to $\text{argmax}_j \text{cosim}(\mathcal{E}(q), \mathcal{E}(\psi^{(j)}))$, where $q$ is the query generated by the core LLM (based on the reasoning) as the action input to action step $a_m$ for an arbitrary iteration $m$ for knowledge retrieval. The retrieved document chunks are summarized (for compression) using an LLM (may be the same as the core LLM or an independent LLM) and are used to update the observation $o_m$. The definition, characteristics, and detection methods for various attacks recorded in the retrieved chunks will facilitate `IDS-Agent` to better understand the potential risks while aggregating the classification results for the ML models. For example, if an attack type can potentially lead to catastrophic results, `IDS-Agent` will be more sensitive to it when any classifier makes such a prediction.

## 4 EXPERIMENTS

### 4.1 EXPERIMENT SETTINGS

**Dataset.** This paper focuses on intrusion detection in the IoT environment, which presents more complexities and challenges than traditional networks. We consider the following two datasets commonly used in previous works.

1) ACI-IoT'23 (Bastian et al., 2023): This dataset contains both benign and malicious network traffic captured from a variety of IoT devices. The dataset includes simulations of several attack types, such as Reconnaissance (e.g., Host Discovery, OS Scan, Ping Sweep, and Port Scan), DoS (e.g., ICMP Flood, SYN Flood, UDP Flood, and Slowloris) and Brute Force (e.g., Dictionary Attacks). We randomly select 10% of the data to train the ML-models for classification. For evaluation, we construct a test dataset from the remaining samples in ACI-IoT'23 by randomly selecting 200 benign samples and 20 samples per attack category.

2) CIC-IoT 2023 (Neto et al., 2023): This dataset simulates large-scale, real-time IoT environments and comprises 33 distinct types of attacks, which is even more difficult for intrusion detection than ACI-IoT'23. The dataset includes network traffic from a broad IoT topology with 105 devices. From the dataset, we selected the 24 most common attack types along with benign samples to create our training and testing datasets. The remaining 9 attack types were excluded from the training data and designated as unknown attacks for evaluating zero-day attack detection performance. Again, we use 10% of the data for training the machine learning classifiers. The test dataset is constructed with 100 benign samples and 10 samples for each attack type.

**Evaluation Metrics.** We are interested in the performance of `IDS-Agent` in both the binary classification of benign and malicious flows and the multi-classification that also requires recognizing the specific attack type when a flow is deemed malicious. For binary classification, we use *accuracy*, and *false alarm rate* (FAR) as the evaluation metrics. Accuracy is the ratio of correctly predicted samples to the total number of samples in the dataset, measuring the overall effectiveness of the `IDS-Agent` in detecting both benign and malicious flows. FAR measures the proportion of false positives (benign flow incorrectly classified as malicious).

For *multi-classification*, we treat each attack category (and the benign category) as a class. We use the per-class *precision*, *recall*, and *F1-score* as the evaluation metrics. Detailed results for the detection of each attack category are deferred to the appendix due to space limitations. In the main paper, we

report the *macro-averaged* precision, recall, and F1-score across all classes as the overall performance of the IDS being evaluated. This macro-average is computed by averaging the metric over all classes with equal weights.

## 4.2 IMPLEMENTATION DETAILS

**The core LLM of `IDS-Agent`.** In our experiments, we consider three LLM choices: GPT-3.5-Turbo, GPT-4o-mini, GPT-4o.

**Tool Design.** 1) **Data preprocessing tool**. We implement Python functions that will be called sequentially for data preprocessing. First, we remove from each flow the fields irrelevant to the traffic features, including the label, time-stamp, and flow ID. Second, we encode the non-numeral fields into numbers, including the connection type and protocol type. Third, we conduct feature selection based on an F-test for linear dependency between features and labels. Finally, the extracted features are standardized. 2) **Classification tool**. The core of the classification tool is an ML model for intrusion detection, which can be self-trained or an off-the-shelf model trained by a third party. Here, we pretrained six (multi-)classifiers, including random forest (RF), K-Nearest Neighbors (KNN), logistic regression (LR), decision tree (DT), multi-layer perceptrons (MLP), and support vector classifier (SVC) using the training set from our benchmarks. The output of the classification tool is the top-3 label predictions with their confidence scores. 3) **Knowledge retrieval tool**. We construct a knowledge base for various IoT attacks by collecting 50 online blogs and 50 research papers. These documents are then split into chunks of 1000 tokens with an overlap of 200 tokens. These chunks were embedded using the OpenAI encoder and stored in a vector database powered by ChromaDB.

## 4.3 BASELINES

1) **Machine learning methods.** We compare `IDS-Agent` with the state-of-the-art ML-based IDS, which uses a quantum-annealing method for feature selection (Davis et al., 2024). In Appendix A.3, we also compare our `IDS-Agent` with six IDSs using the six ML models we pretrained, respectively.

2) **LLM-based methods**. We compared our `IDS-Agent` with the latest GPT-4-based intrusion detection approach (Zhang et al., 2024), which leverages the model's reasoning capabilities and in-context demonstrations. The authors particularly demonstrate that providing GPT-4 with a few labeled examples can improve the accuracy of intrusion detection. We further improve the performance of this baseline by first clustering their in-context examples using a Gaussian Mixture Model (GMM) and then selecting in-context examples from different clusters to cover as many attack cases as possible. Compared to this baseline with a fixed set of demonstrations, `IDS-Agent` retrieves LTM for demonstration dynamically based on input similarity.

3) **Ensemble learning methods (majority vote).** We create a strong baseline by ensemble the results from the six ML models we pretrained through majority voting.

## 4.4 EXPERIMENT RESULTS

**Quantitative Analysis.** The quantitative results on the ACI-IoT'23 dataset are summarized in Table 1. Our `IDS-Agent` achieves the best general performance when GPT-4o is used as the core LLM. Comparable performance is achieved when GPT-4o-mini is used as the core LLM, which is over 60% more cost-effective than GPT-3.5 Turbo. `IDS-Agent` also shows a clearly better recall than the baselines. In particular, when detecting UDP flood attacks, `IDS-Agent` achieves a recall of 0.80 compared with 0.20 for the LLM baseline (GPT-4o) and 0.55 for the majority voting baseline. We also found that for some attack types such as DNS-Flood, Slowloris, and Dictionary attacks, `IDS-Agent` demonstrates slightly lower precision than majority voting. This decrease in precision is due to the `IDS-Agent`'s heightened sensitivity to high-threat attacks, leading it to classify a sample as an attack even with less than 50% voting from the classifiers involved.

It is worth noting that the vanilla GPT-4-based method (Zhang et al., 2024), despite enhancements through in-context learning, performs unsatisfactorily. Moreover, when there are more attack categories, more in-context demonstrations will be required (to effectively inform all attack categories) for their method, posing a significant challenge due to GPT-4's limited token input length and the

Table 1: Binary-classification and multi-classification performance of `IDS-Agent` compared with baseline approaches on the ACI-IoT'23 and CIC-IoT'23 datasets. We compare our method with vanilla GPT-4o-based method enhanced by in-context learning (Zhang et al., 2024), a Random Forest classifier that uses a quantum-annealing method for feature selection (Davis et al., 2024), and a strong baseline of majority voting by six ML classifiers.

| Dataset | Metric Types | Metrics | GPT-4o | RF | Majority Vote | IDS-Agent (GPT-3.5) | IDS-Agent (GPT-4o-mini) | IDS-Agent (GPT-4o) |
|---|---|---|---|---|---|---|---|---|
| ACI-IoT'23 | Binary-Class | Binary-Class Accuracy ↑ | 0.721 | 0.890 | 0.960 | 0.954 | 0.963 | **0.965** |
| | | FAR ↓ | 0.497 | 0.060 | **0.020** | 0.050 | 0.041 | 0.030 |
| | Multi-Class | Multi-Class Accuracy ↑ | 0.678 | 0.790 | **0.980** | 0.972 | 0.976 | **0.980** |
| | | Macro Avg Precision ↑ | 0.682 | 0.785 | 0.980 | 0.971 | 0.980 | **0.982** |
| | | Macro Avg Recall ↑ | 0.754 | 0.760 | 0.961 | 0.952 | 0.972 | **0.972** |
| | | Macro Avg F-Score ↑ | 0.682 | 0.750 | 0.962 | 0.923 | 0.974 | **0.975** |
| CIC-IoT'23 | Binary-Class | Binary-Class Accuracy ↑ | 0.750 | 0.825 | 0.882 | 0.876 | 0.894 | **0.904** |
| | | FAR ↓ | 0.144 | 0.050 | 0.040 | 0.050 | **0.030** | **0.030** |
| | Multi-Class | Multi-Class Accuracy ↑ | 0.610 | 0.751 | 0.771 | 0.762 | 0.788 | **0.795** |
| | | Macro Avg Precision ↑ | 0.580 | 0.755 | 0.760 | 0.759 | 0.790 | **0.800** |
| | | Macro Avg Recall ↑ | 0.450 | 0.692 | 0.700 | 0.694 | 0.723 | **0.733** |
| | | Macro Avg F-Score ↑ | 0.510 | 0.680 | 0.699 | 0.700 | 0.722 | **0.750** |

increase in cost. Our method integrates ML models and utilizes RAG to retrieve the most related knowledge from the memories, significantly reducing the token cost.

Table 1 also presents the evaluation results on the CIC-IoT'23 dataset. Powered by both reasoning ability and tool calls, the `IDS-Agent` achieves higher accuracy than the LLM baseline and majority voting method. Moreover, `IDS-Agent` achieves high detection accuracy on some very challenging attacks, such as ArpSpoofing and Host Discovery. The detailed results for each attack type are deferred to the Appendix A.3.

**Case study** While the majority voting baseline shows relatively strong performance, their classification results usually lack interpretability (Yang et al., 2022). In contrast, we leverages the reasoning capabilities of LLMs to enhance interpretability which may help to improve the detection performance. One example is illustrated on the right of Figure 1 where the `IDS-Agent` concludes that the MITM ARP-Spoofing classification is more likely, as it appears in three classifiers' top-3 predictions with significant confidence in general. Another case for the decision-making is shown in Figure 2, with the ground truth label being 'reconnaissance activities'. Despite 3 out of 6 machine learning models predicting the traffic as benign, `IDS-Agent` accurately labels it as a reconnaissance attack. This decision is rooted in the `IDS-Agent`'s understanding that both Host Discovery and OS Scanning belong to reconnaissance attacks. By considering the relationship between these types of attacks, the `IDS-Agent` demonstrates its ability to detect subtle patterns that individual classifiers may overlook, thus improving the overall detection accuracy. Additional examples are provided in the appendix to demonstrate its decision-making process further.

### 4.5 ZERO-DAY ATTACK

We find that `IDS-Agent` not only excels in detecting known attacks but also shows promise in identifying zero-day threats. Table 2 presents the detection results on the CIC-IoT'23 dataset, comparing `IDS-Agent` with two recent zero-day intrusion detection methods: ACGAN (Zhao et al., 2022) and RealNVP (Matejek et al., 2024). We reproduced ACGAN using its official code and RealNVP based on the FrEIA framework, training separate normalizing flows for benign and malicious samples. We set the detection threshold of RealNVP at the 90th percentile of the negative log-likelihood from the INN model; samples exceeding this threshold are considered unknown attacks or out-of-distribution (OOD). We selected nine attack types from the CIC-IoT'23 dataset as 'unseen' for evaluation, excluding them from training data. Each attack type was evaluated with 50 samples. `IDS-Agent`'s classifiers were trained on data from other known attack categories. Additionally, we enhanced the system prompt with specific instructions to classify ambiguous samples as "Unknown"

**Thought**: I have obtained the predictions from multiple classifiers. Here are the results:
1. **Random Forest**: - **Recon-HostDiscovery**: 0.2800, MITM-ArpSpoofing: 0.2500, BenignTraffic: 0.2400
2. **K-Nearest Neighbors**: **BenignTraffic**: 0.4000, Recon-HostDiscovery: 0.4000, MITM-ArpSpoofing: 0.2000
3. **Logistic Regression**: **Recon-OSScan**: 0.3909, Recon-PortScan: 0.2442, Recon-HostDiscovery: 0.1493
4. **MLP**: **BenignTraffic**: 0.5555, MITM-ArpSpoofing: 0.3421, Recon-OSScan: 0.0586
5. **Support Vector Classifier**: **BenignTraffic**: 0.6324, Recon-HostDiscovery: 0.2326, Recon-OSScan: 0.0855
6. **Decision Tree**: **Recon-HostDiscovery**: 1.0000, Recon-PortScan: 0.0000, DDoS-UDP_Flood: 0.0000

Now, I will analyze the results to determine the final classification. The predictions show a mix of benign and attack classifications,... I will search for more information regarding the features of the attacks and benign traffic to make a more informed decision.
**Action**: knowledge_retriever
**Action Input**: {"query": "what is Recon-HostDiscovery and how to detect it"}
**Observation:** The provided context does not specifically mention \"Recon-HostDiscovery\" or provide details on what it is or how to detect it. Based on general knowledge, "Recon-HostDiscovery" typically refers to the reconnaissance phase in cybersecurity where an attacker attempts to discover hosts on a network. This phase involves scanning the network to identify active devices, open ports, and services running on those devices...
**Final Answer:**
{
"line_number": 229,
 "analysis": "The traffic features were classified by multiple models with discrepancies. The Random Forest and Decision Tree models indicated Recon-HostDiscovery, while Logistic Regression suggested Recon-OSScan. The K-Nearest Neighbors, MLP, and Support Vector Classifier models leaned towards benign traffic. **Given the context of the features and the nature of the predictions, the final classification leans towards reconnaissance activities.**",
"predicted_label_top_1": "Recon-HostDiscovery",
"predicted_label_top_2": "Recon-OSScan",
"predicted_label_top_3": "BenignTraffic",
}

Figure 2: A case study of the `IDS-Agent` detecting reconnaissance activities. Despite most machine learning models classifying the traffic as benign, the `IDS-Agent` ultimately predicts the final label as a reconnaissance attack. This decision is based on its understanding that both Host Discovery and OS Scan belong to reconnaissance activities.

if most classifiers' confidence was below a certain threshold. We set the threshold as 0.7 in our experiments. The recall for `IDS-Agent` in predicting an 'unknown attack' from these unseen classes was measured, achieving a top-1 recall of 0.61. Notably, `IDS-Agent` showed high recall for Vulnerability Scan and SQL Injection attacks, likely due to their distinct deviation from the training data distribution, resulting in low classification confidence. This indicates `IDS-Agent`'s capability to detect unknown attacks, particularly when diverse machine learning models yield divergent results with low confidence or when traffic features are anomalous. The recall for benign examples decreased from 0.91 to 0.86 after introducing the zero-day detection prompt. This drop is attributed to the tendency of the model to classify OOD benign examples as unknown attacks.

Table 2: Comparisons of the recall of different zero-day attack detection methods.

| Methods | Backdoor | DNS Spoofing | Uploading Attack | XSS | Dictionary BruteForce | Command Injection | VulScan | Browser Hijacking | SQL | Avg Recall |
|---|---|---|---|---|---|---|---|---|---|---|
| ACGAN | 0.38 | 0.35 | 0.59 | 0.32 | 0.36 | 0.42 | 0.88 | 0.05 | 0.38 | 0.41 |
| RealNVP | 0.45 | 0.37 | 0.68 | 0.49 | 0.42 | 0.45 | 0.89 | 0.03 | 0.45 | 0.47 |
| IDS-Agent | 0.64 | 0.46 | 0.74 | 0.65 | 0.45 | 0.55 | 0.95 | 0.15 | 0.86 | 0.61 |

## 4.6 ABLATION STUDY

We conduct an ablation study focusing on two key modules: the Knowledge Retrieval Module and the Long-Term Memory Module. We evaluate how each module affects the detection performance on two settings: 1) 'in-distribution' setting with all attack labels known and 2) zero-day attack setting in Section 4.5 with a subset of unknown attacks. We also evaluate the performance under different detection sensitivities.

**Effect of Knowledge Retrieval Module** The Knowledge Retrieval Module is designed to augment the classifiers' outputs with relevant information from an external knowledge base, enhancing the agent's understanding of complex attack patterns. To assess its impact, we disable this module and compare the performance of the modified agent with the original `IDS-Agent`. Table 3 summarizes the detection performance with and without the Knowledge Retrieval Module. When the module is removed, we observe significant decrements in the detection accuracy for both settings. Specifically, the recall for detecting zero-day attacks drops from 0.61 to 0.42. This indicates that without access to

external knowledge, the agent has a diminished ability to recognize patterns that are not represented in the training data.

Table 3: Effect of Knowledge Retrieval Module

| Recall | With KRM | Without KRM |
|---|---|---|
| In-Distribution | 0.733 | 0.710 |
| Zero-Day | 0.610 | 0.420 |

Table 4: Effect of Long-Term Memory Module

| Recall | With LMM | Without LMM |
|---|---|---|
| In-Distribution | 0.733 | 0.702 |
| Zero-Day | 0.610 | 0.560 |

**Effect of Long-Term Memory Module** The Long-Term Memory Module allows `IDS-Agent` to maintain a history of previous observations and decisions, which is essential for detecting attacks exhibiting temporal dependencies and utilizing previous success experiences. To evaluate its effect, we disable the Long-Term Memory Module and assess the agent's performance on the same evaluation set. We set the $\lambda_1$ and $\lambda_2$ as 0.5 in Eq. 1 to balance the recency similarity to retrieve the most relevant past examples from the agent's LTM.

As presented in Table 4, the removal of the Long-Term Memory Module leads to a degradation in detection performance, particularly for attacks that unfold over time, such as Brute Force and Distributed Denial of Service (DDoS) attacks. The overall detection accuracy decreases from 0.733 to 0.702, and the recall for zero-day attacks drops to 0.56. The decrease in performance underscores the importance of the Long-Term Memory Module in capturing temporal features and improving the agent's ability to detect attacks that evolve over time. By retaining historical information, `IDS-Agent` can identify suspicious patterns that may not be apparent when considering individual events in isolation.

**Detection Sensitivity.** Sensitive configuration is a critical function of intrusion detection systems (IDS). In signature-based IDS, experts need to manually adjust detection sensitivity, which can be both costly and time-consuming (Díaz-Verdejo et al., 2022). In contrast, the detection sensitivity of our `IDS-Agent` can be easily adjusted through input prompts. Here, we can optionally instruct the `IDS-Agent` to operate under three different sensitivity levels: *aggressive*, *balanced*, and *conservative*. This sensitivity level controls the trade-off between the false alarm rate and the missed detection rate. The results are shown in Table 5. For the aggressive detection, the `IDS-Agent` achieves high recall for attacks (0.97) but lower recall for benign examples (0.90). Conversely, conservative detection shows relatively lower recall for attacks (0.85) but higher recall for benign examples (0.98). The complete prompts and detection results are shown in Appendix A.4.

Table 5: The classification results of different detection sensitivities. The core LLM is GPT-4o. We compute the precison, recall and F1-score under different sensitivities on the ACI-IoT'23 dataset.

| Sensitivity | Aggressive | | | Balance | | | Conservative | | |
|---|---|---|---|---|---|---|---|---|---|
| Metrics | Precision | Recall | F1-score | Precision | Recall | F1-score | Precision | Recall | F1-score |
| Benign | 0.96 | 0.90 | 0.92 | 0.87 | 0.96 | 0.91 | 0.60 | 0.98 | 0.75 |
| Attack (Macro Avg) | 0.97 | 0.97 | 0.97 | 0.98 | 0.95 | 0.96 | 0.95 | 0.85 | 0.87 |

## 5 CONCLUSION

In this paper, we propose `IDS-Agent`, the first LLM-powered agent for intrusion detection. We design an iterative reasoning-followed-by-action pipeline for `IDS-Agent` to extract data from the network traffic, preprocess the data, consult different machine learning models for classification results and details, retrieve both internal and external knowledge, and summarize the final detection inference. These agent actions are facilitated by a memory module and a wide array of tools for intrusion detection and general purposes. Empirically, `IDS-Agent` outperforms diverse types of SOTA IDSs on ACI-IoT'23 and CIC-IoT'23. We find that when classifiers produce discrepancy predictions, `IDS-Agent` can utilize inherent and external knowledge to assist decision-making. Moreover, `IDS-Agent` can be easily adapted to detect zero-day attacks, exhibiting better performance than existing methods. Finally, we find that `IDS-Agent` effectively follows the sensitivity instructions in detection without requiring expert intervention or additional tuning.

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

# A   APPENDIX A

## A.1   GENERAL PROMPT AND REASONING TRACE OF IDS-AGENT

The general prompt of the `IDS-Agent` is illustrated in Figure 3. The process begins with instructing the `IDS-Agent` to load network traffic data and perform feature preprocessing. Afterward, we utilize a range of classifiers to analyze the data. To enhance decision-making, the `IDS-Agent` retrieves prior successful examples from its knowledge base for comparison. In cases where discrepancies arise between the predictions of different models, we prompt the `IDS-Agent` to consult internal or external knowledge bases for additional insights to resolve the conflict. Finally, the `IDS-Agent` consolidates the findings and presents the result in a structured JSON format. Figure 4 provides an example of the reasoning trace produced by the `IDS-Agent` during this process.

## A.2   ADDITIONAL CASE STUDIES

The cases in Figure 5 and Figure 6 highlight the enhanced reasoning ability of `IDS-Agent` with the knowledge retriever. It is shown that `IDS-Agent` not only considers the top-1 predictions but also the second and third predictions and their confidences. Moreover, in these examples, when the models have discrepancies in the predictions, the `IDS-Agent` automatically accesses external databases to extract additional knowledge, aiding in feature analysis and supporting its final decision. By leveraging these external knowledge sources, the `IDS-Agent` gains a deeper understanding of complex attack patterns and anomalies, enhancing both accuracy and decision-making. This dynamic capability allows the `IDS-Agent` to better adapt to new or evolving threats in the IoT environment.

**General Prompt:**
You are a helpful assistant that can implement multi-step tasks, such as intrusion detection. I will give you the traffic features, you are asked to classify it using tools. The final output must be in a JSON format according to the classifier results. You should plan first such as:
1. Load the traffic features from the CSV file. You can use the load_data_line tool to obtain the complete traffic.
2. Preprocessing the feature. This can be done using the data_preprocessing tool. Input the traffic in the original format.
3. load classifiers for classification. This can be done using the classifier tool. You can use multiple classifiers. The tool params include a classifier name, which must be one from {model_names} and the preprocessed features.
4. Retrieve previous successful reasonings to help you predict. This can be done using the memory_retrieve tool with the classifier's names and their classification results as input.
5. When there are discrepancies/disagreements for different models, you can search from vector database/google/wiki to get more information about the difference of attacks to help you make decisions.
6. At the end, you should summarize the results from these classifiers and provide a final result. Summarize the classification with Balance sensitivity, which means balancing the false alarm rate and the missing alarm rate. The predicted label should be the original format of classifier prediction. The final output format **must** be:
Final Answer:
```json
{ 'line_number': \line_number,
'analysis': str, \here is the Analysis,
'predicted_label_top_1': str,
'predicted_label_top_2': str,
'predicted_label_top_3': str,
}```
**User Input:**
Now, classify the traffic from file name {file_name} with index {line_number}

Figure 3: General prompt and user input

**Thought:** I need to follow the plan to classify the traffic features from line number 230. I will start by loading the traffic features from the CSV file.
**Action:** load_data_line
**Action Input:** {"line_number": 230}
**Observation:** flow_duration: 4.198554983, Header_Length: 10051.49, ...
**Thought:** I have successfully loaded the traffic features from line number 230. Now, I will proceed to preprocess the loaded traffic features to prepare them for classification.
**Action:** data_preprocessing
**Action Input:** {"traffic_features": "flow_duration: 4.198554983, Header_Length: 10051.49,..."}
**Observation:** [-0.355, -0.3088, ...]
**Thought:** I have successfully preprocessed the traffic features. Now, I will proceed to classify the preprocessed features using multiple classifiers to determine if the traffic record is an attack.
**Action:** classifier
**Action Input:** {"modelname": "Random Forest", "traffic_features": [-0.355, -0.3088, ...]}
**Observation:** Top predictions: DDoS-SlowLoris: 0.9100, DDoS-HTTP_Flood: 0.0600, DoS-HTTP_Flood: 0.0200
**Thought:** I have obtained the classification results from the Random Forest model. Now, I will classify the same preprocessed features using additional classifiers to gather more predictions.
...

**Thought:** I have obtained the classification results from all five classifiers. Here are the predictions:
...
Now, I will summarize the results and determine the final predicted labels based on the highest confidence scores from the classifiers.
### Summary of Predictions:
- **DDoS-SlowLoris**: Predicted by Random Forest (0.9100), K-Nearest Neighbors (1.0000), and MLP (0.8543).
- **DDoS-HTTP_Flood**: Predicted by Logistic Regression (0.2586), MLP (0.1033), and Support Vector Classifier (0.6331).
- **Recon-PortScan**: Predicted by Logistic Regression (0.1300) and K-Nearest Neighbors (0.0000).
- **BenignTraffic**: Predicted by Support Vector Classifier (0.2320).
### Final Decision:
The most consistent prediction across the classifiers is **DDoS-SlowLoris**, which received the highest confidence scores from multiple models.
Now, I will prepare the final output in the required JSON format.

**Final Answer:**
```json
{
    "line_number": 230,
    "analysis": "The traffic features indicate a high likelihood of a DDoS-SlowLoris attack based on the predictions from multiple classifiers, particularly the Random Forest and K-Nearest Neighbors models.",
    "predicted_label_top_1": "DDoS-SlowLoris",
    "predicted_label_top_2": "DDoS-HTTP_Flood",
    "predicted_label_top_3": "Recon-PortScan"
}
```

Figure 4: An example of the reasoning trace and final answer.

## A.3 THE PERFORMANCE OF DIFFERENT ML CLASSIFIERS

Table 6 shows the F-score of different ML classifiers on the ACI-IoT'23 dataset as well as our method. Among the six classifiers, MLP achieves the highest F-score of 0.96. The `IDS-Agent` outperforms all six classifiers and the majority vote method. Moreover, we achieve a high F-score on the UDP Flood attack, while the majority method only has an F-score of 0.55.

Table 7 shows the F-score of different ML classifiers on the CIC-IoT'23 dataset as well as our method. Among the six classifiers, Random Forest achieves the highest F-score of 0.75. For the `IDS-Agent`, we use the GPT-4o as the core LLM. It is shown that our attack achieved a higher F-score compared with the majority vote classifier. Moreover, we achieved a higher F-score on the benign traffic compared with six classifiers and the majority vote method, which means our method

**Thought**: I have obtained predictions from all classifiers. Now, I will summarize the results to determine the final classification. The predictions from the classifiers are as follows:
- Random Forest: DDoS-SYN_Flood (0.6458)
- K-Nearest Neighbors: DoS-SYN_Flood (1.0000)
- Logistic Regression: DDoS-SYN_Flood (0.6473)
- MLP: DDoS-SYN_Flood (0.7261)
- Support Vector Classifier: DDoS-SYN_Flood (0.6329)
- Decision Tree: DDoS-SYN_Flood (0.6736)

The majority of classifiers predict "DDoS-SYN_Flood" as the top label. However, K-Nearest Neighbors predicts "DoS-SYN_Flood" with a certainty of 1.0000. To resolve this discrepancy, I will search for more information regarding the features of DDoS and DoS attacks to understand the differences better.

**Action:** knowledge_retriever
**Action Input:** {"query": "What are the features of DoS and DDoS attacks and their differences?"}
**Observation:** DoS (Denial-of-Service) and DDoS (Distributed Denial-of-Service) attacks have several features and key differences:
**Features of DoS Attacks:**...
**Features of DDoS Attacks:**...
**Key Differences:**
- **Source of Attack:** DoS attacks come from one source, while DDoS attacks come from multiple sources.
- **Volume of Traffic:** DDoS attacks typically generate a larger volume of traffic, making them more damaging.
- **Detection and Mitigation:** DDoS attacks are harder to detect and eliminate quickly due to their distributed nature, whereas DoS attacks can be easier to manage since they originate from a single source.
Overall, DDoS attacks are generally considered more severe and challenging to defend against compared to DoS attacks.
**Final Answer:**
{
"line_number": 83,
"analysis": "The majority of classifiers predict 'DDoS-SYN_Flood', indicating a distributed denial-of-service attack. The K-Nearest Neighbors classifier predicts 'DoS-SYN_Flood' with high certainty, but given the nature of the features and the consensus among other classifiers, 'DDoS-SYN_Flood' is deemed the more appropriate classification.",
"predicted_label_top_1": "DDoS-SYN_Flood",
"predicted_label_top_2": "DoS-SYN_Flood",
"predicted_label_top_3": "DDoS-SynonymousIP_Flood"
}

Figure 5: Additional case study on the DDoS attacks of the `IDS-Agent`. From the final analysis, it is shown that the `IDS-Agent` not only considers the predicted labels but also considers the confidence of different classifiers.

**Thought**: I have obtained the predictions from multiple classifiers. Here are the results:
1. **Random Forest**: - **MITM-ArpSpoofing**: 0.4200 - BenignTraffic: 0.2500 - Recon-PortScan: 0.1300
2. **K-Nearest Neighbors**: - **Recon-PortScan**: 0.4000 - MITM-ArpSpoofing: 0.4000 - BenignTraffic: 0.2000
3. **Logistic Regression**: - **BenignTraffic**: 0.3689 - MITM-ArpSpoofing: 0.3105 - DoS-UDP_Flood: 0.1391
4. **MLP**: - **MITM-ArpSpoofing**: 0.4019 - BenignTraffic: 0.3296 - Recon-HostDiscovery: 0.1095
5. **Support Vector Classifier**: - **BenignTraffic**: 0.6327 - MITM-ArpSpoofing: 0.2328 - Recon-HostDiscovery: 0.0855
6. **Decision Tree**: - **Recon-OSScan**: 1.0000 - Recon-PortScan: 0.0000 - DDoS-UDP_Flood: 0.0000
Now, I will analyze the results to determine the final classification. The predictions show a mix of benign and attack classifications, with some classifiers leaning towards "MITM-ArpSpoofing" and others towards "BenignTraffic." The Decision Tree classifier is particularly confident in classifying it as "Recon-OSScan."To resolve the discrepancies, I will search for more information regarding the features of the attacks and benign traffic to make a more informed decision.
**Action**: knowledge_retriever
**Action Input:** {"query": "what is the features of MITM-ArpSpoofing and how to detect it"}
**Observation:** The features of MITM-ARP-Spoofing include: 1. **Interception of Communication**: The attacker positions themselves between two network devices, allowing them to intercept and potentially modify the data being transmitted.
2. **ARP Cache Poisoning**: The attacker sends falsified ARP messages to both devices, associating their MAC address with the IP address of the other device, thus redirecting traffic through their machine....
**Final Answer:**
{
"line_number": 83,
"analysis": "The traffic features indicate a potential MITM-ArpSpoofing attack based on the predictions from multiple classifiers, particularly Random Forest and MLP. The Decision Tree classifier also suggests a reconnaissance attack, but the overall consensus leans towards MITM.",
"predicted_label_top_1": "MITM-ArpSpoofing",
"predicted_label_top_2": "BenignTraffic",
"predicted_label_top_3": "Recon-PortScan",
}

Figure 6: Additional case study on the MITM attacks of the `IDS-Agent`. From the final analysis, it is shown that the `IDS-Agent` not only considers the top 1 prediction but also considers the second and third predictions.

has a lower false alarm rate, which is an important metric for intrusion detection. Figure 7 shows the confusion matrix of the majority voting classifier and `IDS-Agent`.

Table 6: The F-score of different ML classifiers on the ACI-IoT'23 dataset. For the `IDS-Agent`, we use the GPT-4o as the core LLM.

| Model | RF | LR | KNN | MLP | DT | SVC | Majority Vote | IDS-Agent |
|-------|----|----|-----|-----|----|----|---------------|-----------|
| Benign | 0.90 | 0.59 | 0.91 | 0.91 | 0.91 | 0.80 | 0.91 | 0.91 |
| DNS Flood | 0.95 | 0.10 | 0.80 | 0.95 | 0.91 | 0.91 | 1.00 | 0.95 |
| Dictionary Attack | 1.00 | 0.71 | 0.98 | 0.95 | 1.00 | 0.92 | 1.00 | 1.00 |
| ICMP Flood | 1.00 | 0.98 | 0.98 | 1.00 | 0.95 | 0.98 | 0.98 | 0.98 |
| OS Scan | 1.00 | 1.00 | 1.00 | 1.00 | 1.00 | 1.00 | 1.00 | 1.00 |
| Ping Sweep | 0.98 | 0.98 | 0.97 | 0.98 | 0.97 | 0.98 | 1.00 | 1.00 |
| Port Scan | 1.00 | 1.00 | 1.00 | 1.00 | 1.00 | 1.00 | 1.00 | 1.00 |
| SYN Flood | 1.00 | 1.00 | 1.00 | 1.00 | 0.98 | 1.00 | 1.00 | 1.00 |
| Slowloris | 1.00 | 0.43 | 1.00 | 1.00 | 1.00 | 0.97 | 1.00 | 1.00 |
| UDP Flood | 0.60 | 0.00 | 0.45 | 0.74 | 0.50 | 0.00 | 0.55 | 0.80 |
| Vulnerability Scan | 1.00 | 1.00 | 1.00 | 1.00 | 1.00 | 1.00 | 1.00 | 1.00 |
| **Macro Avg** | **0.95** | **0.71** | **0.92** | **0.96** | **0.93** | **0.87** | **0.96** | **0.97** |

Table 7: The F-score of different ML classifiers on the CIC-IoT'23 dataset. For the `IDS-Agent`, we use the GPT-4o as the core LLM.

| Model | DT | KNN | LR | MLP | RF | SVC | Majority Vote | IDS-Agent |
|-------|----|----|----|-----|----|----|---------------|-----------|
| BenignTraffic | 0.79 | 0.77 | 0.79 | 0.75 | 0.75 | 0.73 | 0.74 | 0.84 |
| DDoS-ACK_Fragmentation | 0.98 | 0.95 | 0.95 | 0.93 | 0.95 | 0.98 | 0.95 | 1.00 |
| DDoS-HTTP_Flood | 0.58 | 0.53 | 0.24 | 0.79 | 0.68 | 0.38 | 0.69 | 0.70 |
| DDoS-ICMP_Flood | 0.98 | 0.95 | 0.98 | 0.95 | 1.00 | 1.00 | 1.00 | 1.00 |
| DDoS-ICMP_Fragmentation | 1.00 | 1.00 | 1.00 | 1.00 | 1.00 | 1.00 | 1.00 | 1.00 |
| DDoS-PSHACK_Flood | 1.00 | 1.00 | 0.98 | 1.00 | 1.00 | 1.00 | 1.00 | 0.95 |
| DDoS-RSTFINFlood | 1.00 | 1.00 | 1.00 | 1.00 | 1.00 | 1.00 | 1.00 | 1.00 |
| DDoS-SYN_Flood | 0.72 | 0.08 | 0.63 | 0.73 | 0.76 | 0.64 | 0.75 | 0.75 |
| DDoS-SlowLoris | 0.76 | 0.74 | 0.00 | 0.89 | 0.79 | 0.44 | 0.79 | 0.82 |
| DDoS-SynonymousIP_Flood | 0.70 | 0.74 | 0.70 | 0.72 | 0.74 | 0.65 | 0.74 | 0.78 |
| DDoS-TCP_Flood | 0.69 | 0.08 | 0.66 | 0.67 | 0.69 | 0.67 | 0.67 | 0.71 |
| DDoS-UDP_Flood | 0.71 | 0.23 | 0.77 | 0.75 | 0.71 | 0.74 | 0.71 | 0.74 |
| DDoS-UDP_Fragmentation | 0.97 | 0.95 | 0.95 | 0.97 | 0.97 | 0.97 | 0.97 | 0.95 |
| DoS-HTTP_Flood | 0.65 | 0.79 | 0.73 | 0.85 | 0.76 | 0.89 | 0.83 | 0.84 |
| DoS-SYN_Flood | 0.62 | 0.57 | 0.31 | 0.72 | 0.76 | 0.37 | 0.72 | 0.80 |
| DoS-TCP_Flood | 0.32 | 0.60 | 0.00 | 0.00 | 0.32 | 0.00 | 0.17 | 0.33 |
| DoS-UDP_Flood | 0.53 | 0.65 | 0.55 | 0.52 | 0.48 | 0.46 | 0.48 | 0.46 |
| MITM-ArpSpoofing | 0.54 | 0.58 | 0.09 | 0.60 | 0.58 | 0.58 | 0.62 | 0.67 |
| Mirai-greeth_flood | 0.95 | 0.98 | 0.90 | 0.97 | 0.97 | 1.00 | 0.97 | 0.95 |
| Mirai-greip_flood | 0.98 | 0.97 | 0.89 | 0.98 | 0.98 | 1.00 | 0.98 | 0.95 |
| Mirai-udpplain | 0.95 | 0.98 | 1.00 | 0.98 | 0.98 | 1.00 | 0.98 | 1.00 |
| Recon-HostDiscovery | 0.54 | 0.45 | 0.40 | 0.48 | 0.55 | 0.39 | 0.47 | 0.53 |
| Recon-OSScan | 0.30 | 0.30 | 0.08 | 0.00 | 0.15 | 0.10 | 0.17 | 0.15 |
| Recon-PortScan | 0.41 | 0.36 | 0.24 | 0.31 | 0.44 | 0.31 | 0.37 | 0.31 |
| **Macro Avg** | **0.74** | **0.68** | **0.62** | **0.73** | **0.75** | **0.68** | **0.70** | **0.75** |

## A.4  DETAILS FOR SENSITIVITY CUSTOMIZATION

We adjust the detection sensitivity by prompting the core LLM with "Summarize the classification with {sensitivity} sensitivity, {sensitivity details}". Here, 'sensitivity details' will be 'discover the attack as the priority', 'balance the false alarm rate and the missing alarm rate', and 'do not alert unless you are very sure', for 'sensitivity' being 'aggressive', 'balanced' and 'conservative'. The detection performances of `IDS-Agent` for different detection sensitivities are shown in Table 8. It is shown that the 'Aggressive' command achieves a higher recall on the attacks while the 'Conservative' command achieves a higher recall on the benign examples. The classification results, detailed in Table 8 of the appendix, show that the `IDS-Agent` effectively follows these sensitivity instructions without requiring expert intervention or additional tuning.

Table 8: The classification results of different detection sensitivities.

| Sensitivity | Aggressive | | | Balance | | | Conservative | | |
|---|---|---|---|---|---|---|---|---|---|
| Metrics | Precision | Recall | F1-score | Precision | Recall | F1-score | Precision | Recall | F1-score |
| Benign | 0.96 | 0.90 | 0.92 | 0.87 | 0.96 | 0.91 | 0.60 | 0.98 | 0.75 |
| DNS Flood | 0.91 | 1.00 | 0.95 | 0.91 | 1.00 | 0.95 | 0.94 | 0.80 | 0.86 |
| Dictionary Attack | 0.91 | 1.00 | 0.95 | 1.00 | 1.00 | 1.00 | 1.00 | 0.65 | 0.79 |
| ICMP Flood | 0.95 | 1.00 | 0.89 | 0.95 | 1.00 | 0.98 | 0.95 | 1.00 | 0.98 |
| OS Scan | 1.00 | 1.00 | 1.00 | 1.00 | 1.00 | 1.00 | 1.00 | 1.00 | 1.00 |
| Ping Sweep | 0.95 | 1.00 | 0.98 | 1.00 | 1.00 | 1.00 | 1.00 | 1.00 | 1.00 |
| Port Scan | 1.00 | 1.00 | 1.00 | 1.00 | 1.00 | 1.00 | 1.00 | 1.00 | 1.00 |
| SYN Flood | 1.00 | 1.00 | 1.00 | 1.00 | 1.00 | 1.00 | 1.00 | 1.00 | 1.00 |
| Slowloris | 0.95 | 1.00 | 0.98 | 1.00 | 1.00 | 1.00 | 1.00 | 0.40 | 0.57 |
| UDP Flood | 1.00 | 0.80 | 0.89 | 1.00 | 0.53 | 0.69 | 1.00 | 0.47 | 0.64 |
| Vulnerability Scan | 1.00 | 1.00 | 1.00 | 1.00 | 1.00 | 1.00 | 1.00 | 1.00 | 1.00 |
| **Macro Avg** | **0.97** | **0.97** | **0.97** | **0.98** | **0.95** | **0.96** | **0.95** | **0.85** | **0.87** |

## A.5  THE ZERO-DAY ATTACK DETECTION DETAILS

We prompt GPT-4o to classify an example as an unknown attack if multiple classifiers output low confidence for their top predictions or if there are conflicting predictions among different classifiers. This is based on our observation that, for unknown attacks, machine learning models typically exhibit relatively low confidence levels, as shown in Figure 8. Specifically, we instruct the LLM to consider an example as a potential unknown attack if more than two models have low confidence (e.g., below a threshold of 0.7). Moreover, if more than two models have low confidence or if different models produce significantly divergent predictions, we direct `IDS-Agent` to search the knowledge base for characteristics of the most probable predicted attacks. If the traffic features do not match these attack characteristics, we confirm the example as an unknown attack and provide this as the final output.

## A.6  THE INFLUENCE OF HYPERPARAMETERS

To assess the influence of different values of $\lambda_1$ and $\lambda_2$ in Eq. 1, we conducted experiments by varying these parameters and measuring the impact on retrieval effectiveness and overall classification performance. Table 9 summarizes the results of our experiments. The experimental results indicate that both recency and content similarity are crucial for effective LTM retrieval. A balanced approach, where $\lambda_1$ and $\lambda_2$ are equal, provides the best performance, suggesting that the agent benefits from considering both embedding similarity and recency.

Table 9: Performance metrics for different values of $\lambda_1$ and $\lambda_2$.

| $\lambda_1$ | $\lambda_2$ | Accuracy (%) | Precision (%) | Recall (%) |
|---|---|---|---|---|
| 0.1 | 0.9 | 97.2 | 97.2 | 96.5 |
| 0.5 | 0.5 | **98.0** | **98.2** | **97.2** |
| 0.9 | 0.1 | 97.3 | 97.1 | 96.1 |

## A.7 EXCUTION TIME OF IDS-AGENT

In this section, we evaluate the execution time of the proposed IDS-Agent and compare it with the in-context-learning-based GPT-4 approach. We conducted the execution time experiments with the Intel Core i7 CPU of 3.8GHz. The operating system is MacOS 14.6. As shown in Table 10, the IDS-Agent balances performance and efficiency, averaging 8.65 seconds per instance. We use GPT-4o API as the core LLM of IDS-Agent. The additional time compared to the GPT-4 method is due to the knowledge retrieval and aggregation process, but it remains well within acceptable limits for real-time applications.

Table 10: Execution time comparison between different methods.

| Method | GPT-4 | IDS-Agent |
|---|---|---|
| Average Time per Instance (s) | 3.36 | 8.65 |

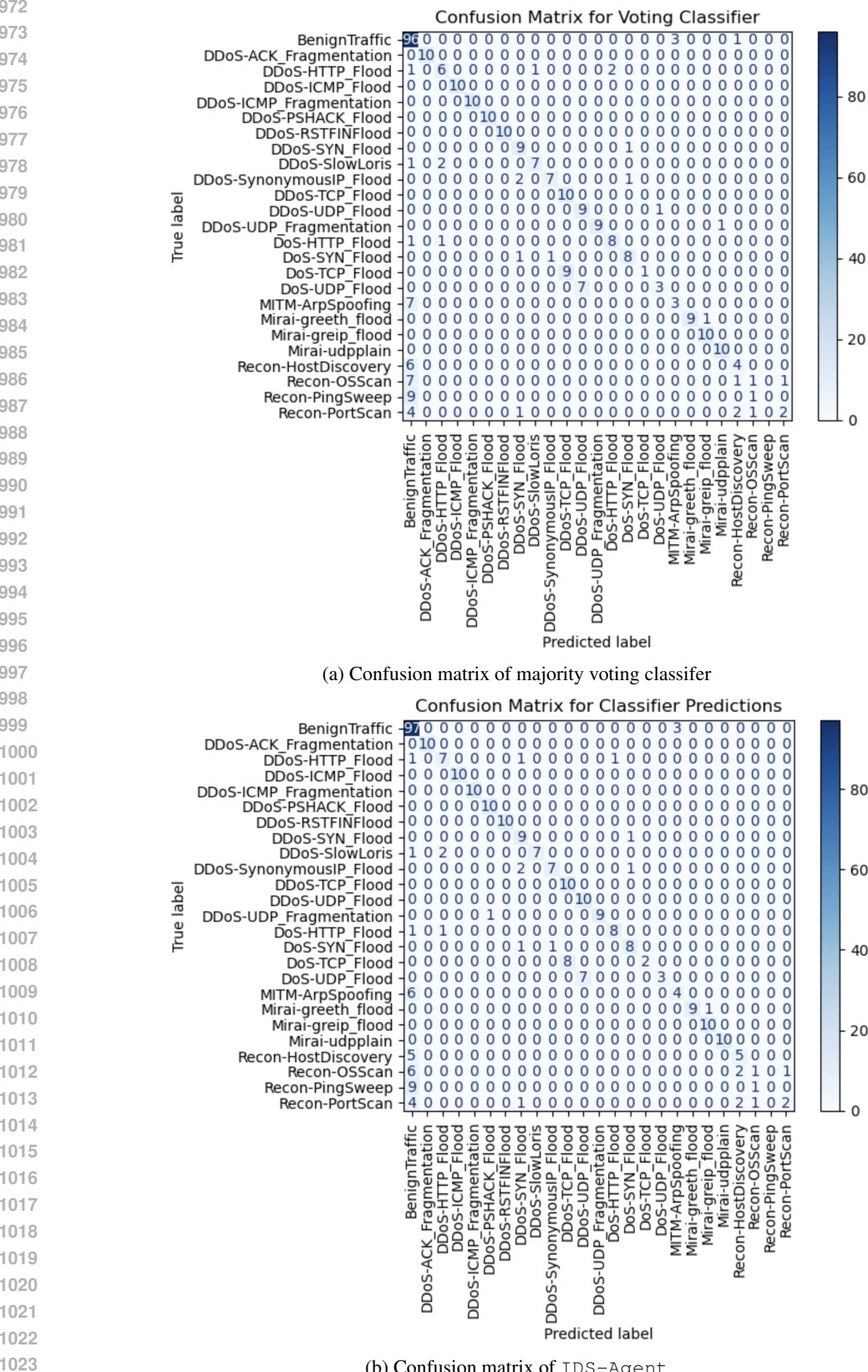

(a) Confusion matrix of majority voting classifer

(b) Confusion matrix of `IDS-Agent`

Figure 7: The confusion matrix of majority voting classifier and `IDS-Agent` on the CIC-IoT'23 dataset.

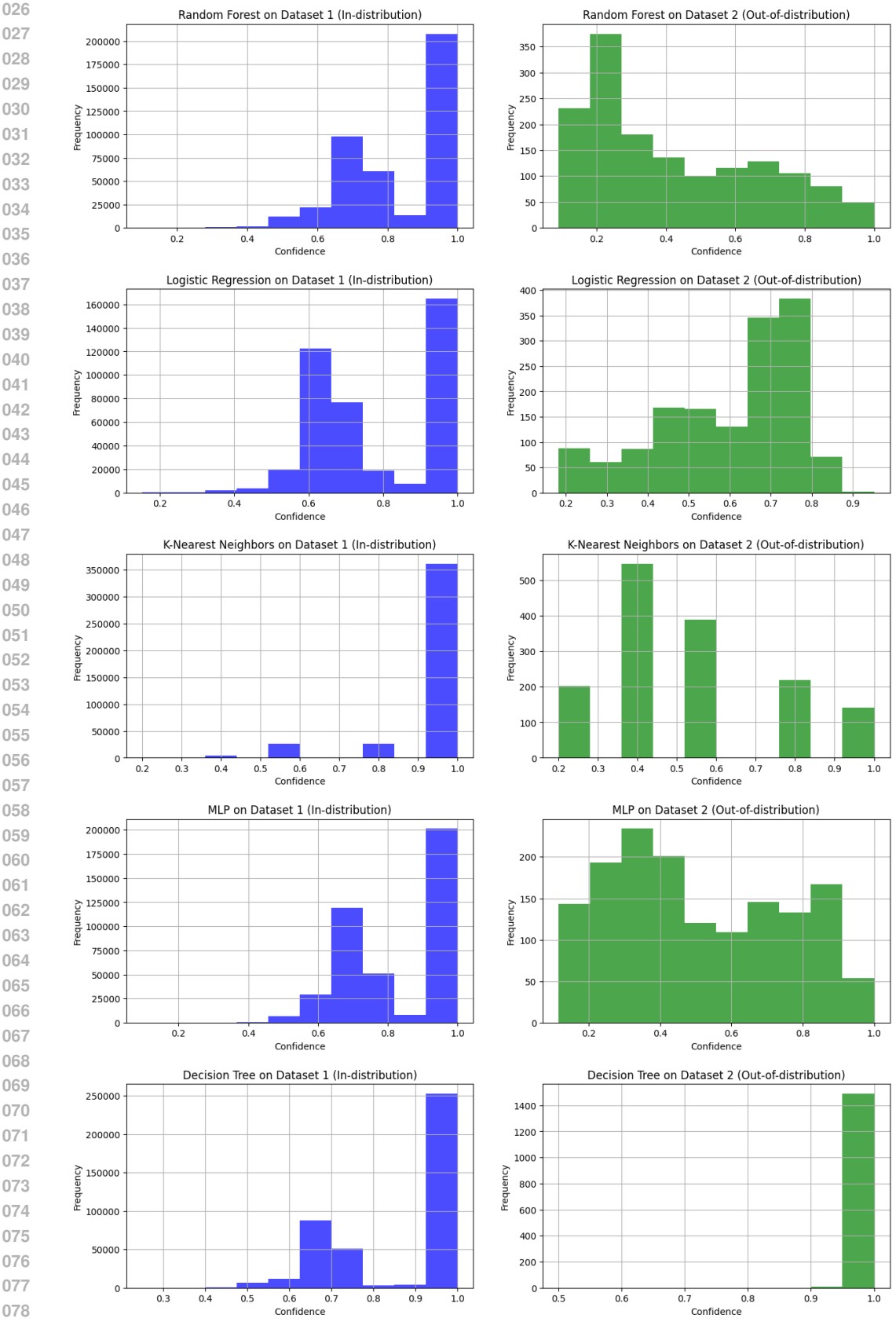

Figure 8: The confidence distributions of difference classifiers on the in-distribution dataset and out-of-distribution dataset.

