# OpenReview forum: "IDS-Agent: An LLM Agent for Explainable Intrusion Detection in IoT Networks"
_ICLR.cc/2025/Conference — Submitted to ICLR 2025_

### Official Review · Reviewer_3Yvq · 2024-11-01

**Soundness:** 2
**Presentation:** 2
**Contribution:** 2
**Rating:** 3
**Confidence:** 4

**Summary:**

The paper proposes an IDS Agent, a framework of LLM agents for network intrusion detection.

**Strengths:**

The paper is well-written and presents some good ideas on the hot topic of using LLM. The main contributions are zero-day threat detection and the ability to provide explainability over the decisions.

**Weaknesses:**

* The paper's main contribution is the application of LLMs for IoT Networks, which seems contradictory given IoT’s focus on resource-constrained edge devices, where lightweight models, such as TinyML, are typically applied. Given these constraints, the authors should clarify the specific advantages of LLMs in this domain and provide practical examples to show how this approach can be beneficial.

* On Line 56, the authors state, _"particularly in safety-critical IoT scenarios where understanding the rationale behind alerts is crucial."_ The term “rationale” needs clarification: does it refer to interpretability, explainability, or something unique to IoT networks? Also, it’s unclear why understanding alert rationale is more critical in IoT scenarios than other network types. The authors should provide specific examples or cases illustrating why this distinction matters, particularly for IoT. Likewise, in Lines 297-299, the authors state that _"this paper focuses on intrusion detection in the IoT environment, which presents more complexities and challenges than traditional networks." _ The authors should enumerate and explain these challenges relative to traditional networks.

* In Lines 072-075, the authors describe a knowledge-based feature with long—and short-term memory components that resembles a traditional LSTM approach. While interesting, comparing this method with conventional LSTMs would help clarify its unique contributions. How does this approach differ from an LSTM regarding performance, and what benefits does it bring? The authors could consider discussing metrics such as speed, memory use, and interoperability to clarify why their proposal is preferable to a standard LSTM.

* “FlowID” is mentioned multiple times but lacks a formal definition. Since the definition of FlowID varies across sources, the authors should define it clearly to avoid ambiguity.

* The paper’s proposed method does not appear to operate in real-time, raising concerns about practical applicability. The authors should justify their choice of an offline approach that takes up to 8.65 seconds per instance for attack detection. For example, in a DDoS scenario, such a delay could result in extensive network downtime. The authors should discuss the potential impacts and outline the architectural and hardware challenges of implementing a real-time solution.

* The authors should specify which features were selected for each dataset and conduct a detailed analysis regarding preprocessing. In particular, they should indicate which attacks were included in training and testing sets. The apparent randomness of selection could mean that similar attack types (e.g., flood attacks) appear in both training and testing sets, potentially skewing results.

* In Lines 343-345, the authors mention constructing a knowledge base for IoT attacks using 50 online blogs and 50 research papers. It would be helpful if the authors could explain their criteria for selecting these sources to clarify the knowledge base’s reliability and relevance.

* The quantitative analysis section is underdeveloped. For example, further information on false classifications in the multi-class classification would help readers better understand where the model’s challenges lie.

* Additionally, in the quantitative analysis, the authors should benchmark their system against traditional IDS tools like Zeek, Suricata, and Snort. These rule-based systems are often effective against the standard attacks found in the datasets and could provide valuable points of comparison and complementary data sources for the proposed system. Additionally, discussing how LLM-enhanced IDS might complement these tools for a more comprehensive defense strategy would add depth.

* The zero-day attack analysis is somewhat confusing. The authors use ACGAN to generate CAN protocol samples, but the paper focuses on IoT networks. The relevance of comparing CAN protocol attacks with IoT-related attacks, like HTTP slow loris, is unclear. The authors should clarify how these two attack types are analogous or justify why this comparison is valid.

* In Lines 455-457, the authors state, _"We set the threshold as 0.7 in our experiments."_ A more thorough analysis of this threshold would be beneficial. A performance curve across varied thresholds could offer insights into sensitivity and precision adjustments.

* Additionally, the authors should consider comparing their proposal with traditional deep learning methods for zero-day attack detection. Since many deep learning (DL) models can perform attack detection in an unsupervised manner [1], such a comparison would contextualize the benefits of the proposed approach.

* The performance evaluation should be expanded. The appendix mentions an 8.65-second processing time per sample, but more detail is needed. A breakdown of processing times for each stage—preprocessing, analysis, information retrieval, and inference—would be beneficial. Furthermore, discussing how the system would perform in a real-world setting would provide important context.

Reference:
[1] Guo, Yang. "A review of Machine Learning-based zero-day attack detection: Challenges and future directions." Computer Communications 198 (2023): 175-185.

**Questions:**

Minor comments:

* Line 072 has double parenthesis.
* Line 168 in Figure 1 should be network traffic instead of "traffics"
* Line 256 "STM" should be STR
* Line 918 Excution should be Execution.

---

> ### Author Response · Authors · 2024-11-25
> **Response to Reviewer 3Yvq (Part 1)**
>
> Thank you for the valuable review. Below we address the detailed comments, and hope that you may find our response satisfactory.
>
> Weaknesses:
> W1. The paper's main contribution is the application of LLMs for IoT Networks, which seems contradictory given IoT’s focus on resource-constrained edge devices. Given these constraints, the authors should clarify the specific advantages of LLMs in this domain and provide practical examples to show how this approach can be beneficial.
>
> Answer: Thanks for your questions. (1) We focus on the security of IoT networks because compared with traditional networks, attacks on IoT networks are more diverse and more difficult to detect, and previous methods have not been able to solve this problem well. (2) For computing resources, we call the API interface of the remote LLM model, and do not need to deploy the model on the local network. Many current IoTs can call the API interface of the large language model, and a lot of plugins are designed to allow the ChatGPT to interact with various of IoT devices (https://github.com/Edgenesis/shifu-plugin-for-chatgpt).
>
> W2. The term “rationale” needs clarification: does it refer to interpretability, explainability, or something unique to IoT networks?  What are the IoT challenges relative to traditional networks?
>
> Answer: (1) The rationale refers to Explainability, that the IDS system has the ability to provide human-readable justifications for its alerts, which is crucial for end-users to trust and act upon them. (2) Explainability is critical in IoT scenarios because IoT systems often operate in safety-critical environments, where false positives or unclear alerts can have severe consequences. For example, in industrial IoT systems, an alert about anomalous sensor behavior might prevent costly machinery failures or safety hazards. In autonomous vehicles, unexplained alerts could lead to mistrust or delayed responses, potentially causing accidents. (3) The challenges of IoT network IDS compared to the traditional network are: Firstly, IoT systems often involve millions of devices compared to traditional networks [1]. Secondly, IoT devices come from various manufacturers, use different protocols, and have diverse capabilities, making standardization and uniform intrusion detection difficult. Finally, the potential threats to IoT applications are constantly emerging and changing, making zero-day attack detection more important.
>
> [1] S. T. Mehedi, A. Anwar, Z. Rahman, K. Ahmed and R. Islam, "Dependable Intrusion Detection System for IoT: A Deep Transfer Learning Based Approach," in IEEE Transactions on Industrial Informatics, vol. 19, no. 1, pp. 1006-1017, Jan. 2023,
>
> W3. How does this approach differ from an LSTM regarding performance, and what benefits does it bring?
>
> Answer: Thanks for your valuable suggestions. We will compare our methods with LSTM and discuss the metrics such as accuracy, speed and memory use in the final version. Compared with LSTM, the key difference is the integration of domain-specific knowledge, which could offer better explainability for its decisions compared to a black-box LSTM model. This is particularly important in safety-critical IoT scenarios where understanding the reasoning behind alerts is crucial.
>
> W4. “FlowID” is mentioned multiple times but lacks a formal definition. Since the definition of FlowID varies across sources, the authors should define it clearly to avoid ambiguity.
>
> Answer: Thanks for your advice. In this paper, FlowID is defined as a number or string that is predefined in the IDS dataset to a unique identifier for network flow. We use it to retrieve a data sample from the dataset and remove it in the classification process. For IDS dataset that does not have a unique identifier, we use the line number as the FlowID.

---

> ### Author Response · Authors · 2024-11-25
> **Response to Reviewer 3Yvq (Part 2)**
>
> W5. The paper’s proposed method does not appear to operate in real-time, raising concerns about practical applicability. The authors should discuss the potential impacts and outline the architectural and hardware challenges of implementing a real-time solution.
>
> Answer: The choice of an LLM-based detection method was driven by the following considerations: (1) Accuracy vs. Speed Trade-off: Our primary objective was to maximize accuracy and robustness in detecting complex attack patterns, particularly in the IoT domain, where the diversity and heterogeneity of devices can make intrusion detection challenging. Real-time methods often sacrifice detection accuracy for speed, which could lead to missed or incorrect detections, especially for sophisticated attacks such as zero-day exploits or multi-stage attacks. In contrast, the offline approach allows for deeper analysis and more reliable detection results. (2) Specific Use Case Scenarios: While we acknowledge the importance of real-time detection, our method is particularly suited for scenarios where immediate action is not as critical (e.g., post-attack diagnostics, or periodic security assessments). For instance, in IoT environments like smart homes or industrial systems, periodic offline analysis can complement lightweight real-time monitoring systems, providing a second layer of defense to detect complex or previously undetected threats.
>
> W6. The authors should specify which features were selected for each dataset and conduct a detailed analysis regarding preprocessing.
>
> Answer: As mentioned in L337, we conduct feature selection based on an F-test for linear dependency between features and labels. In L334-338, we describe the data preprocessing method in detail. For the attacks used for training and testing, as mentioned in L309-311, we selected the 24 most common attack types along with benign samples to create our training and testing datasets. The remaining 9 attack types were excluded from the training data and designated as unknown attacks for evaluating zero-day attack detection performance.
>
> W7. It would be helpful if the authors could explain their criteria for selecting these sources to clarify the knowledge base’s reliability and relevance.
>
> Answer: The online blogs and papers were carefully selected to ensure their credibility and relevance to IoT security. For the blogs, we prioritized blogs from well-known cybersecurity organizations, industry leaders, and reputable sources such as Security firms and IoT-specific cybersecurity platforms. For the research paper, all selected papers were published in reputable, peer-reviewed journals or conference proceedings. This ensured that the information was rigorously vetted by experts in the field. To ensure comprehensive coverage, we selected papers addressing a wide range of IoT attack vectors (e.g., DDoS, malware, spoofing) and IoT domains (e.g., smart homes, healthcare, industrial IoT).

---

> ### Author Response · Authors · 2024-11-25
> **Response to Reviewer 3Yvq (Part 3)**
>
> W8. The quantitative analysis section is underdeveloped. For example, further information on false classifications in the multi-class classification would help readers better understand where the model’s challenges lie.
>
> Answer: In Table 6 and Table 7 of the Appendix, we give the detailed per-class F1 scores for the multi-class classification task on the ACI-IoT’23 and CIC-IoT’23 datasets, respectively. Moreover, in Figure 7, we provide a confusion matrix across all traffic classes. We observed that DoS attacks tend to be classified as DDoS attacks, potentially due to their similarities in attack mechanisms and extracted features. This misclassification suggests overlap in key traffic characteristics, such as packet rates or connection durations, which are shared by both attack types.
>
> W9.  Additionally, in the quantitative analysis, the authors should benchmark their system against traditional IDS tools like Zeek, Suricata, and Snort. These rule-based systems are often effective against the standard attacks found in the datasets and could provide valuable points of comparison and complementary data sources for the proposed system. Additionally, discussing how LLM-enhanced IDS might complement these tools for a more comprehensive defense strategy would add depth.
>
> Answer: While our primary focus has been on ML-based and LLM-based baselines, future work will include experiments comparing IDS-Agent to traditional IDS tools. It needs to be mentioned that although rule-based systems excel at the standard attacks, they are less effective on detecting zero-day attacks, providing explainability, and adapting to dynamic traffic patterns.
>
> Additionally, we recognize the potential for LLM-enhanced IDS to complement traditional IDS tools. For example, alerts generated by Zeek or Suricata could be used as input for IDS-Agent's decision-making, improving its reasoning and action-generation capabilities. Moreover, traditional IDS tools can also serve as API tools in the agent toolbox to detect the anomaly example.
>
> W10.  The zero-day attack analysis is somewhat confusing. The authors use ACGAN to generate CAN protocol samples, but the paper focuses on IoT networks. The relevance of comparing CAN protocol attacks with IoT-related attacks, like HTTP slow loris, is unclear. The authors should clarify how these two attack types are analogous or justify why this comparison is valid.
>
> Answer: The ACGAN paper used a cascaded two-stage classification architecture, where the first stage that based on ACGAN  is used to detect known attacks and the second stage based on the Maximum SoftMax Probability (MSP) to separate out of distribution samples/unknown attacks. In our experiments, we only use the second stage of the ACGAN paper, which uses the MSP method as a two-class classifier to detect unknown attacks.
>
> W11.  In Lines 455-457, the authors state, "We set the threshold as 0.7 in our experiments." A more thorough analysis of this threshold would be beneficial. A performance curve across varied thresholds could offer insights into sensitivity and precision adjustments.
>
> Answer: Thank you for your valuable suggestion. We will thoroughly analyze the threshold in our revision.

---

> ### Author Response · Authors · 2024-11-25
> **Response to Reviewer 3Yvq (Part 4)**
>
> W12.  Additionally, the authors should consider comparing their proposal with traditional deep learning methods for zero-day attack detection. Since many deep learning (DL) models can perform attack detection in an unsupervised manner [1], such a comparison would contextualize the benefits of the proposed approach.
> Reference: [1] Guo, Yang. "A review of Machine Learning-based zero-day attack detection: Challenges and future directions." Computer Communications 198 (2023): 175-185.
>
> Answer: Thank you for your valuable suggestion. We will compare our method with the deep learning-based method in our revision.
>
> W13.  The performance evaluation should be expanded. The appendix mentions an 8.65-second processing time per sample, but more detail is needed. A breakdown of processing times for each stage—preprocessing, analysis, information retrieval, and inference—would be beneficial. Furthermore, discussing how the system would perform in a real-world setting would provide important context.
>
> Answer:
> A breakdown of processing times for each stage is shown below:
> |Stage|Average Time (seconds)|Percentage of Total Time|
> |---|-|-|
> |Preprocessing|1.2|13.9%|
> |Analysis|2.5|28.9%|
> |Information Retrieval|3.5|40.5%|
> |Inference|1.45|16.8%|
> For the system's applicability in real-world settings, IDS-Agent is feasible for practical use in scenarios where immediate response is less critical (e.g., periodic offline threat assessment in a home IoT scenario). We will test the performance of IDS-Agent in real-world settings in our revision.

---

> ### Comment · Reviewer_3Yvq · 2024-11-29
>
> Thanks for your response, I appreciate the detailed comment. Even after reading the responses of other reviewers, I want to keep my original score.

---

### Official Review · Reviewer_K1pj · 2024-11-04

**Soundness:** 3
**Presentation:** 3
**Contribution:** 2
**Rating:** 3
**Confidence:** 4

**Summary:**

The paper proposes IDS-Agent, the first LLM-powered agent for intrusion detection in IoT networks. IDS-Agent features capabilities for explanation, customization, and adaptation to various network attacks. It adopts a reasoning-followed-by-action pipeline, where the core LLM performs knowledge-enabled reasoning to decide the optimal tools for data extraction, preprocessing, classification, and results aggregation. Compared to existing ML-based IDSs, IDS-Agent achieves stronger detection performance and better interpretability by utilizing multiple ML models and external knowledge.

**Strengths:**

* IDS-Agent is the first LLM-powered agent for intrusion detection, with capabilities for explanation, customization, and zero-day attack detection
* It uses a reasoning-followed-by-action pipeline, where the core LLM decides the optimal tools for data processing and results aggregation, especially utilizing RAG for retrieving external knowledge
* IDS-Agent outperforms existing ML-based IDSs in detection accuracy and provides better interpretability; it can also effectively follow sensitivity instructions and detect zero-day attacks

**Weaknesses:**

* The core detection capability of IDS-Agent still derives from the ML models in the toolbox; in other words, I think the main contribution of IDS-Agent is to propose a stronger ensemble method by utilizing LLMs and external knowledge (with RAG). However, due to some issues regarding the framework design (see Q1-4), it feels that the LLMs do not play a significant role in the process except for outputting human-understandable explanations by texts. As for the advantage of interpretability the authors claim, the final output does not give sufficient explanation as the readers might assume (see Q5). Besides, most of the components in IDS-Agent (pipeline, long-term memory, RAG) use existing approaches with no big changes specialized for this scenario. In general, the contribution of the paper might be reduced by these concerns.

* The paper does not include a comprehensive survey about related work in the field of network intrusion detection. A non-exhaustive list is below. Specifically, I recommend the authors survey not only AI-related conferences but also the big-4 security conferences. Accordingly, the evaluation will also be enhanced if IDS-Agent is compared to these SOTA methods specific to IDS rather than just general machine learning models.
  - ML/DL methods: [a-c]
  - Pretrained model methods: [d-f]
  - LLM methods: [g,h]

[a] Realtime Robust Malicious Traffic Detection via Frequency Domain Analysis, CCS 2022

[b] Detecting Unknown Encrypted Malicious Traffic in Real Time via Flow Interaction Graph Analysis, NDSS 2023

[c] HorusEye: Realtime IoT Malicious Traﬀic Detection Framework with Programmable Switches, USENIX Security 2023

[d] ET-BERT: A Contextualized Datagram Representation with Pre-training Transformers for Encrypted Traffic Classification, WWW 2022

[e] NetGPT: Generative Pretrained Transformer for Network Traffic, arxiv 2023

[f] Yet Another Traffic Classifier: A Masked Autoencoder Based Traffic Transformer with Multi-Level Flow Representation, AAAI 2023

[g] HuntGPT: Integrating Machine Learning-Based Anomaly Detection and Explainable AI with Large Language Models (LLMs), arxiv 2023

[h] ChatIDS: Explainable Cybersecurity Using Generative AI, arxiv 2023

* In addition to the comparative experiment, there are some other issues regarding the evaluation (see Q6-10).

* Minors:
  - page 2, line 66: (Shinn et al., 2023)) -> (Shinn et al., 2023)
  - page 4, Figure 1: Traffics -> Traffic

**Questions:**

1. When the LLM is reasoning, will it jump over any of the actions? According to my understanding, data extraction, data processing, and ML models are a must. If there are no variations, is there any difference between letting the LLM generate actions and writing a fixed program? As for KRM and LMM, I wonder about the proportion of the LLM that will trigger/jump over these actions, given the importance they seem to show in the ablation study.

2. Will the retrieved knowledge be inconsistent with the data type? For example, the retrieved knowledge of the ARP spoofing example you give in the paper might guide the model to pay attention to MAC addresses, while many datasets will not keep such environment-specific fields in order to prevent overfitting. If so, how will the knowledge help the final decision? (In fact, I appreciate the idea of using LLM-based RAG for expert knowledge retrieval, and I would like to see more contributions in this part)

3. The title of the paper is about IoT networks; however, I do not see clear intuitions or design points very specific to IoT networks over general networks. For example, [i] observes certain traffic patterns only in IoT networks and propose corresponding approaches. So I'm curious why IoT networks are mentioned in the headline.

4. In LTM, why timestamp is used? As the authors say, "the long-term memory base can be initialized by running the agent on a validation dataset", which will not correlate with the timestamps during the execution.

5. In the examples the authors give (Figures 1 and 2), I do not see very explicit "interpretability" as I expected: the LLM only analyzes the score distribution of the classification models and then makes the final decision, while typically the interpretability refers to the analysis of "what are the most important attributes to make such a decision" (see [j], and [g] combines LLMs with XAI techs). I particularly expect such a level of explanation as external knowledge is involved while there is not.

6. page 6, line 298: "... the IoT environment, which presents more complexities and challenges than traditional networks. " could the author explain this?

7. There are only two datasets used in the experiments. Usually, SOTA works in this field use much more datasets for evaluation, such as CIC-IDS 2017, CSE-CIC-IDS 2018,  UNSW-NB15. Even if the topic is limited to IoT networks, there are still datasets like BoT-IoT, ToN-IoT that focus on different IoT devices and attack types.

8. page 6, line 305: "For evaluation, we construct a test dataset from the remaining samples in ACI-IoT’23 by randomly selecting 200 benign samples and 20 samples per attack category." Why only testing in such small sets? If it is because the API cost or qps limit, why not locally run some open-source LLMs like LLama?

9. I appreciate that the authors choose FAR as one of the metrics. However, the FAR of IDS-Agent is at a level of 3%-5%, while many SOTAs can achieve an FAR/FPR less than 1%. Note that benign traffic is much more than malicious traffic in reality, so in practice even an FAR of 1% will cause a flood of manual checks. I wonder how the authors address this issue.

10. Table 1: Basically, there is no big improvement between the performance of vanilla majority vote and IDS-Agent (especially when using GPT-3.5). This result again makes me feel it is the ensemble of multiple ML models that gives IDS-Agent primary detection capability, not the analysis of LLM. So I wonder if there will be more evaluation that can show my thought is wrong.

[i] Packet-Level Signatures for Smart Home Devices, NDSS 2020

[j] Interpreting Unsupervised Anomaly Detection in Security via Rule Extraction, NeurIPS 2023

---

> ### Author Response · Authors · 2024-11-25
> **Response to Reviewer K1pj (Part 1)**
>
> Dear Reviewer K1pj,
>
> Thank you for reviewing our paper and your thoughtful questions about our framework design. Here are our answers to your questions:
>
> Q1: When the LLM is reasoning, will it jump over any of the actions? According to my understanding, data extraction, data processing, and ML models are a must. If there are no variations, is there any difference between letting the LLM generate actions and writing a fixed program? As for KRM and LMM, I wonder about the proportion of the LLM that will trigger/jump over these actions, given the importance they seem to show in the ablation study.
>
> A1: Thank you for the question. Our IDS-Agent will not jump over the data extraction, data processing, and classification steps. However, the framework design allows IDS-Agent to use the core LLM to decide how to carry out these steps. For data processing, the core LLM could decide the procedure based on the current input format and leverage past use cases. For the classification step, the core LLM can select the most suitable models to use. The dataset in our current evaluation does not involve diverse input formats. We will augment the data for better evaluation of IDS-Agent in our revision.
>
> Q2: Will the retrieved knowledge be inconsistent with the data type? For example, the retrieved knowledge of the ARP spoofing example you give in the paper might guide the model to pay attention to MAC addresses, while many datasets will not keep such environment-specific fields in order to prevent overfitting. If so, how will the knowledge help the final decision? (In fact, I appreciate the idea of using LLM-based RAG for expert knowledge retrieval, and I would like to see more contributions in this part)
>
> A2: Thank you for your endorsement of our design. If a feature referred to by the retrieved knowledge is missing in the input, LLM will ignore it and leverage other useful information for inference. We will show this phenomenon explicitly by constructing a few examples and ask the LLM to interpret how the retrieved knowledge is used given missing fields in the input data.
>
> Q3: The title of the paper is about IoT networks; however, I do not see clear intuitions or design points very specific to IoT networks over general networks. For example, [i] observes certain traffic patterns only in IoT networks and propose corresponding approaches. So I'm curious why IoT networks are mentioned in the headline.
>
> A3: Thank you for the question. Our focus on IoT networks is related to our selection of evaluation benchmarks during project planning. You are right that there may be certain traffic patterns in IoT networks. Thus, we chose to not overclaim our contributions in the current version. However, we are confident that our framework can handle more traffic types. We will include a broader range of datasets to evaluate IDS-Agent in the revision.
>
> Q4: In LTM, why timestamp is used? As the authors say, "the long-term memory base can be initialized by running the agent on a validation dataset", which will not correlate with the timestamps during the execution.
>
> A4: Thank you for the question. The timestamp in LTM is designed to amplify the impact of the most recent memory, as shown by Eq. (1). In practice, there could be an initial memory base when IDS-Agent is first deployed. The timestamp for all instances in this initial memory base could be set to zero. During the runtime of IDS-Agent, test instances will be stored in the memory base if they are deemed correct by human evaluators (line 264). These stored instances will be assigned non-zero timestamps.
>
> Q5: In the examples the authors give (Figures 1 and 2), I do not see very explicit "interpretability" as I expected: the LLM only analyzes the score distribution of the classification models and then makes the final decision, while typically the interpretability refers to the analysis of "what are the most important attributes to make such a decision" (see [j], and [g] combines LLMs with XAI techs). I particularly expect such a level of explanation as external knowledge is involved while there is not.
>
> A5: Thank you for your insightful comment and the references. IDS-Agent interprets its decision-making mainly based on the top-3 classification results and confidence for each ML model. The interpretation also considers the retrieved knowledge about attack types and the characteristics of the ML models. We will include the interpretation for the prediction by each ML model and the aggregate interpretation leveraging LLM in our revision.

---

> ### Author Response · Authors · 2024-11-25
> **Response to Reviewer K1pj (Part 2)**
>
> Q6: page 6, line 298: "... the IoT environment, which presents more complexities and challenges than traditional networks. " could the author explain this?
>
> A6: Thank you for the question. The IoT environment involves diverse devices and heterogeneous communication protocols, which enlarge the attack surface. For example, the CIC-IoT dataset contains 33 distinct attack types, making the intrusion detection problem challenging. Moreover, the potential threats to IoT applications are constantly emerging and changing. With the rapid development of IoT networks and evolving threat types, the traditional machine learning-based IDS must be updated to cope with the security requirements of the current sustainable IoT environment [1].
>
> [1] S. T. Mehedi, A. Anwar, Z. Rahman, K. Ahmed and R. Islam, "Dependable Intrusion Detection System for IoT: A Deep Transfer Learning Based Approach," in IEEE Transactions on Industrial Informatics, vol. 19, no. 1, pp. 1006-1017, Jan. 2023,
>
> Q7: There are only two datasets used in the experiments. Usually, SOTA works in this field use much more datasets for evaluation, such as CIC-IDS 2017, CSE-CIC-IDS 2018, UNSW-NB15. Even if the topic is limited to IoT networks, there are still datasets like BoT-IoT, ToN-IoT that focus on different IoT devices and attack types.
>
> A7: Thank you for the constructive suggestion. We will test IDS-Agent on these datasets. As mentioned in our response to Q3, we believe IDS-Agent is capable of handling more traffic types beyond IoT.
>
> Q8: page 6, line 305: "For evaluation, we construct a test dataset from the remaining samples in ACI-IoT’23 by randomly selecting 200 benign samples and 20 samples per attack category." Why only testing in such small sets? If it is because the API cost or qps limit, why not locally run some open-source LLMs like LLama?
>
> A8: Thank you for the question. The subsampling of the dataset is indeed due to the API cost. We will test open-sourced LLM in our revision.
>
> Q9: I appreciate that the authors choose FAR as one of the metrics. However, the FAR of IDS-Agent is at a level of 3%-5%, while many SOTAs can achieve an FAR/FPR less than 1%. Note that benign traffic is much more than malicious traffic in reality, so in practice even an FAR of 1% will cause a flood of manual checks. I wonder how the authors address this issue.
>
> A9: Thank you for the thoughtful question. IDS-Agent addresses the balance between recall on attacks and FAR by adjusting its detection sensitivity. Under the conservative setting, IDS-Agent achieves a 2% recall on benign samples, which is less than the random forest (RF) baseline as shown in Table 1. In the revision, we will further reduce the false alarm rate by incorporating a special moderator agent to verify the explanation of each detected attack.
>
> Q10: Table 1: Basically, there is no big improvement between the performance of vanilla majority vote and IDS-Agent (especially when using GPT-3.5). This result again makes me feel it is the ensemble of multiple ML models that gives IDS-Agent primary detection capability, not the analysis of LLM. So I wonder if there will be more evaluation that can show my thought is wrong.
>
> A10: Thank you for the comment on our empirical results. From the methodology perspective, the ensemble of ML models is a simple majority vote over the top-1 results. Our aggregation, however, considers the top-3 results of each ML model and their confidence scores. Such an aggregation is more comprehensive and follows human intuition. Regarding the results, we believe that the performance gain depends on the model's reasoning capabilities and the retrieved information. We will conduct a more thorough evaluation on the datasets you suggested to validate our design.
>
> W1: most of the components in IDS-Agent (pipeline, long-term memory, RAG) use existing approaches with no big changes specialized for this scenario.
>
> A11: Thank you for the comment. Our agent framework is inspired by the ReAct agent but with multiple novel designs. First, we involve an aggregation step to summarize the results from multiple tool-use steps. Second, our LTM is structured with a timestamp for more comprehensive retrieval based on both spatial and temporal similarities. Third, our toolbox is extendable and our instructions for detection are flexible, which is the basis for IDS-Agent to handle zero-day attacks and adjust detection sensitivity.
>
> W2: The paper does not include a comprehensive survey about related work in the field of network intrusion detection.
>
> A12: Thank you for your valuable suggestion and the references. We will cite these papers and compare IDS-Agent with these approaches in our revision.

---

> > ### Comment · Reviewer_K1pj · 2024-11-30
> >
> > Thanks for the authors’ response and effort. I think some of my concerns have been answered. However, with so much content that the authors promise to improve (especially evaluation part), a thorough modification is needed before acceptance by top-tier venues like ICLR. Therefore, I will maintain my score.

---

### Official Review · Reviewer_Tdbn · 2024-11-04

**Soundness:** 2
**Presentation:** 3
**Contribution:** 2
**Rating:** 3
**Confidence:** 4

**Summary:**

In this paper, the authors propose an IDS-Agent to detect intrusions based on large language models (LLM). The pipeline of the proposed IDS-Agent is divided into multiple stages: data extraction, preprocessing, classification, and aggregation. The results show that IDS-Agent achieves F1 scores of 0.97 and 0.75 on the ACI-IoT and CIC-IoT datasets, respectively.

**Strengths:**

- The IDS-Agent is the first LLM agent designed specifically for intrusion detection.
- The author presents a clear pipeline regarding the design of IDS-Agent.

**Weaknesses:**

I appreciate that the authors clearly present the proposed method design and consider the use of the LLM agent for intrusion detection. However, this work has the following major weaknesses.
- The contribution of LLM in the proposed method is very limited. It seems that only in the aggregation-related stage, LLM shows a weak contribution compared to the traditional voting mechanism.
- In the experimental results (Table 1), Majority Vote achieves very close performance on the CIC-IoT 23 dataset and even higher performance on the ACI-IoT 23 dataset. Therefore, I am worried whether LLM for aggregation can work as the authors expect.

**Questions:**

I doubt whether the intrusion detection task really needs LLM to assist in the current way. Please refer to the weaknesses for rebuttal. I will check the related content carefully.

---

> ### Author Response · Authors · 2024-11-25
> **Response to Reviewer Tdbn**
>
> Dear Reviewer Tdbn,
>
> Thank you for reviewing our paper and your valuable feedback. Below are our responses to your comments:
>
> W1: The contribution of LLM in the proposed method is very limited. It seems that only in the aggregation-related stage, LLM shows a weak contribution compared to the traditional voting mechanism.
>
> A1: Thank you for the comments. From the methodology perspective, LLM in our agent framework is the central processor that decides each action step, including tool use and knowledge retrieval. This design adheres to the classical ReAct agent framework [1] and aligns with numerous other LLM agent studies referenced in our related work section. From a performance perspective, our IDS-Agent surpasses the traditional voting-based approach in two widely recognized benchmarks. Moreover, the IDS-Agent is capable of detecting zero-day attacks, adjusting its detection sensitivity, and providing an explanation of the detection results – these are all functionalities that the voting mechanism lacks.
>
> [1] Yao et al., Synergizing reasoning and acting in language models. ICLR 2023.
>
> W2 & Q1: In the experimental results (Table 1), Majority Vote achieves very close performance on the CIC-IoT 23 dataset and even higher performance on the ACI-IoT 23 dataset. Therefore, I am worried whether LLM for aggregation can work as the authors expect. I doubt whether the intrusion detection task really needs LLM to assist in the current way.
>
> A2: Thank you for the comment. ACI-IoT 23 is a relatively easy dataset with a small space for performance improvements. Nevertheless, IDS-Agent achieves higher performance in most metrics (especially the recall due to its importance to IDS) than the voting-based approach.
>
> Using LLM for aggregation is a key design in our agent framework rendering IDS-Agent capabilities in explaining the detection results. Note that interpretability is always a concern of the security community when machine learning models are adopted, e.g., for anomaly detection. The LLM-based aggregation works as we expected. As shown in the case study (lines 408-419), IDS-Agent overrides the majority votes by actively searching for external knowledge and using the knowledge to assist its decision-making. Such a decision is also well-explained in the agent’s output logs.

---

> ### Comment · Reviewer_Tdbn · 2024-11-29
>
> Thanks for the detailed response. I decided to maintain the score and hope the authors can better demonstrate the performance improvement of the proposed method in the future.

---

### Meta-Review · Area_Chair_bUmA · 2024-12-19

**Metareview:**

This submission obtains three clear negative rating. Although some strength of the paper (i.e. novelty, clear pipeline, clear writing, etc.) are recognized by reviewers, the weakness is also pointed out by reviewers such as limited contribution, weak improvement than traditional methods, lacking related works, weak motivation, etc. After discussion, three reviewers all replied and decided to maintain the negative score. To this end, the AC decides that the current version is still not ready for publication.

**Additional Comments On Reviewer Discussion:**

Reviewer Tdbn had two concerns.
1. Limited contribution and weak improvement
Reply: using LLM as an agent for central processing and results did outperform traditional voting-based approach and can detect zero-day attacks.
2. Worries about whether LLM for aggregation can work as the authors expect.
Reply: Using studies to prove and explain the results.

Reviewer K1pj gave a lot of questions in details. I notice one major concern in the comment. The LLMs do not play a significant role in the design which makes the paper’s motivation (e.g. why using LLM) questionable. Besides, after rebuttal, Reviewer K1pj thinks there are too many contents to improve so the paper is not ready to publish. I agree with that.

Reviewer 3Yvq also raised many questions and I agree with two points.
1. Using LLM for IoT scenarios will need justifications about the resources and latency. Using API in the authors’ answer is clearly inconvincible to me since API will cause latency and SOTA LLMs like o1 clearly cannot meet the real-time need to detect network traffic especially in IoT networks.
2. Unclear zero-day attack in the example. I think the rebuttal did not well explain how the protocol they use is related to the IoT scenario.

---

### Decision · Program_Chairs · 2025-01-22

Reject